# Variable Importance using Decision Trees

**S. Jalil Kazemitabar**
UCLA
sjalilk@ucla.edu

**Arash A. Amini**
UCLA
aaamini@ucla.edu

**Adam Bloniarz**
UC Berkeley*
adam@stat.berkeley.edu

**Ameet Talwalkar**
CMU
talwalkar@cmu.edu

## Abstract

Decision trees and random forests are well established models that not only offer good predictive performance, but also provide rich feature importance information. While practitioners often employ variable importance methods that rely on this impurity-based information, these methods remain poorly characterized from a theoretical perspective. We provide novel insights into the performance of these methods by deriving finite sample performance guarantees in a high-dimensional setting under various modeling assumptions. We further demonstrate the effectiveness of these impurity-based methods via an extensive set of simulations.

## 1   Introduction

Known for their accuracy and robustness, decision trees and random forests have long been a workhorse in machine learning [1]. In addition to their strong predictive accuracy, they are equipped with measures of variable importance that are widely used in applications where model interpretability is paramount. Importance scores are used for model selection: predictors with high-ranking scores may be chosen for further investigation, or for building a more parsimonious model.

One common approach naturally couples the model training process with feature selection [2, 5]. This approach, which we call TREEWEIGHT, calculates the feature importance score for a variable by summing the impurity reductions over all nodes in the tree where a split was made on that variable, with impurity reductions weighted to account for the size of the node. For ensembles, these quantities are averaged over constituent trees. TREEWEIGHT is particularly attractive because it can be calculated without any additional computational expense above the standard training procedure.

However, as the training procedure in random forests combines several complex ingredients—bagging, random selection of predictor subsets at nodes, line search for optimal impurity reduction, recursive partitioning—theoretical investigation into TREEWEIGHT is extremely challenging. We propose a new method called DSTUMP that is inspired by TREEWEIGHT but is more amenable to analysis. DSTUMP assigns variable importance as the impurity reduction at the root node of a single tree.

In this work we characterize the finite sample performance of DSTUMP under an additive regression model, which also yields novel results for variable selection under a linear model, both with correlated and uncorrelated design. We corroborate our theoretical analyses with extensive simulations in which we evaluate DSTUMP and TREEWEIGHT on the task of feature selection under various modeling assumptions. We also compare the performance of these techniques against established methods whose behaviors have been theoretically characterized, including Lasso, SIS, and SpAM [12, 3, 9].

Our work provides the first finite-sample high-dimensional analyses of tree-based variable selection techniques, which are commonly used in practice but lacking in theoretical grounding. Although we focus on DSTUMP, which is a relatively simple tree-based variable selection approach, our novel proof techniques are highly non-trivial and suggest a path forward for studying more general multi-level tree-based techniques such as TREEWEIGHT. Moreover, our simulations demonstrate that such algorithmic generalizations exhibit impressive performance relative to competing methods under more realistic models, e.g., non-linear models with interaction terms and correlated design.

## 2 Related Work

Our analysis is distinct from existing work in analyzing variable importance measures of trees and forests in several ways. To our knowledge, ours is the first analysis to consider the high-dimensional setting, where the number of variables, $p$, and the size of the active set $s$, grow with the sample size $n$, and potentially $p \gg n$.

The closest related work is the analysis of [8], which considers a fixed set of variables, in the limit of infinite data ($n = \infty$). Unlike DSTUMP's use of the root node only, [8] does consider importance scores derived from the full set of splits in a tree as in TREEWEIGHT. However, they make crucial simplifying (and unrealistic) assumptions that are distinct from those of our analysis: (1) each variable is split on only once in any given path from the root to a leaf of the tree; (2) at each node a variable is picked uniformly at random among those not yet used at the parent nodes, i.e., the splits themselves are not driven by impurity reduction; and (3) all predictors are categorical, with splits being made on all possible levels of a variable, i.e., the number of child nodes equals the cardinality of the variable being split. Our analysis instead considers continuous-valued predictors, the split is based on actual impurity reduction, and our results are nonasymptotic, i.e. they give high-probability bounds on impurity measures for active and inactive variables that hold in finite samples.

A second line of related work is motivated by a permutation-based importance method [1] for feature selection. In practice, this method is computationally expensive as it determines variable importance by comparing the predictive accuracy of a forest before and after random permutation of a predictor. Additionally, due to the algorithmic complexity of the procedure, it is not immediately amenable to theoretical analysis, though the asymptotic properties of a simplified variant of the procedure have been studied in [6].

While our work is the first investigation of finite-sample model selection performance of *tree-based* regression methods, alternative methods performing both linear and nonparametric regression in high dimensions have been studied in the literature. Considering model selection consistency results, most of the attention has been focused on the linear setting, whereas the nonparametric (nonlinear) setup has been mostly studied in terms of the prediction consistency. Under a high-dimensional linear regression model, LASSO has be extensively studied and is shown to be minimax optimal for variable selection under appropriate regularity conditions, including the uncorrelated design with a moderate $\beta_{\min}$ condition. Remarkably, while not tailored to the linear setting, we show that DSTUMP is nearly minimax optimal for variable selection in the same uncorrelated design setting (cf. Corollary 1). In fact, DSTUMP can be considered a nonlinear version of SIS [4], itself a simplified form of the LASSO when one ignores correlation among features (cf. Section 3 for more details).

The Rodeo framework [7] performs automatic bandwidth selection and variable selection for local linear smoothers, and is tailored to a more general nonparametric model with arbitrary interactions. It was shown to possess model selection consistency in high dimensions; however, the results are asymptotic and focus on achieving optimal prediction rate. In particular, there is no clear $\beta_{\min}$ threshold as a function of $n$, $s$, and $p$. RODEO is also computationally burdensome for even modest-sized problems (we thus omit it our experimental results in Section 4).

Among the nonlinear methods, SPAM is perhaps the most well-understood in terms of model selection properties. Under a general high-dimensional sparse additive model, SPAM possesses the *sparsistency* property (a term for model selection consistency); the analysis is reduced to a linear setting by considering expansions in basis functions, and selection consistency is proved under an *irrepresentible* condition on the coefficients in those bases. We show that DSTUMP is model selection consistent in the sparse additive model with uncorrelated design. Compared to SPAM results, our conditions are stated directly in terms of underlying functions and are not tied to a particular basis;

hence our proof technique is quite different. There is no implicit reduction to a linear setting via basis expansions. Empirically, we show that DSTUMP indeed succeeds in the settings our theory predicts.

## 3 Selection consistency

The general model selection problem for non-parametric regression can be stated as follows: we observe noisy samples $y_i = f(x_{i1}, \ldots, x_{ip}) + w_i$, $i = 1, \ldots, n$ where $\{w_i\}$ is an i.i.d. noise sequence. Here, $p$ is the total number of features (or covariates) and $n$ is the total number of observations (or the sample size). In general, $f$ belongs to a class $\mathcal{F}$ of functions from $\mathbb{R}^p \to \mathbb{R}$. One further assumes that the functions in $\mathcal{F}$ depend on at most $s$ of the features, usually with $s \ll p$. That is, for every $f \in \mathcal{F}$, there is some $f_0 : \mathbb{R}^s \to \mathbb{R}$ and a subset $S \subset [p]$ with $|S| \leq s$ such that $f(z_1, \ldots, z_p) = f_0(z_S)$ where $z_S = (z_i, i \in S)$. The subset $S$, i.e., the set of *active features*, is unknown in advance and the goal of model selection is to recover it given $\{(y_i, x_i)\}_{i=1}^n$. The problem is especially challenging in the high-dimensional setting where $p \gg n$. We will consider various special cases of this general model when we analyze DSTUMP. For theoretical analysis it is common to assume $s$ to be known and we will make this assumption throughout. In practice, one often considers $s$ to be a tunable parameter that can be selected, e.g., via cross-validation or greedy forward selection.

We characterize the model selection performance of DSTUMP by establishing its *sample complexity*: that is, the scaling of $n$, $p$, and $s$ that is sufficient to guarantee that DSTUMP identifies the active set of features with probability converging to 1. Our general results, proved in the technical report, allow for a correlated design matrix and additive nonlinearities in the true regression function. Our results for the linear case, derived as a special case of the general theory, allow us to compare the performance of DSTUMP to the information theoretic limits for sample complexity established in [11], and to the performance of existing methods more tailored to this setting, such as the Lasso [12].

Given a generative model and the restriction of DSTUMP to using root-level impurity reduction, the general thrust of our result is straightforward: impurity reduction due to active variables concentrates at a significantly higher level than that of inactive variables. However, there are significant technical challenges in establishing this result, mainly deriving from the fact that the splitting procedure renders the data in the child nodes non-i.i.d., and hence standard concentration inequalities do not immediately apply. We leverage the fact that the DSTUMP procedure considers splits at the median of a predictor. Given this median point, the data in each child node is i.i.d., and hence we can apply standard concentration inequalities in this conditional distribution. Removing this conditioning presents an additional technical subtlety. For ease of exposition, we first present our results for the linear setting in Section 3.1, and subsequently summarize our general results in Section 3.2. We provide a proof of our result in the linear setting in Section 3.3, and defer the proof of our general result to the supplementary material.

---

**Algorithm 1** DSTUMP
___
**input** $\{x_k \in \mathbb{R}^n\}_{k=1}^{k=p}, y \in \mathbb{R}^n$, # top features $s$
  $m = \frac{n}{2}$
  **for** $k = 1, \ldots, p$ **do**
    $\mathcal{I}(x_k) = \texttt{SortFeatureValues}(x_k)$
    $y^k = \texttt{SortLabelByFeature}(y, \mathcal{I}(x_k))$
    $y_{[m]}^k, y_{[n] \setminus [m]}^k = \texttt{SplitAtMidpoint}(y^k)$
    $i_k = \texttt{ComputeImpurity}(y_{[m]}^k, y_{[n] \setminus [m]}^k)$
  **end for**
  $\mathcal{S} = \texttt{FindTopImpurityReductions}(\{i_k\}, s)$
**output** top $s$ features sorted by impurity reduction

---

**The DSTUMP algorithm.** In order to describe DSTUMP more precisely, let us introduce some notation. We write $[n] := \{1, \ldots, n\}$. Throughout, $y = (y_i, i \in [n]) \in \mathbb{R}^n$ will be the response vector observed for a sample of size $n$. For an ordered index set $\mathcal{I} = (i_1, i_2, \ldots, i_r)$, we set $y_{\mathcal{I}} = (y_{i_1}, y_{i_2} \ldots, y_{i_r})$. A similar notation is used for unordered index sets. We write $x_j = (x_{1j}, x_{2j}, \ldots, x_{nj}) \in \mathbb{R}^n$ for the vector collecting values of the $j$th feature; $x_j$ forms the $j$th column of the *design matrix* $X \in \mathbb{R}^{n \times p}$.

Let $\mathcal{I}(x_j) := (i_1, i_2 \ldots, i_n)$ be an ordering of $[n]$ such that $x_{i_1 j} \leq x_{i_2 j} \leq \cdots \leq x_{i_n j}$ and let $\text{sor}(y, x_j) := y_{\mathcal{I}(x_j)} \in \mathbb{R}^n$; this is an operator that sorts $y$ relative to $x_j$. DSTUMP proceeds as

follows: Evaluate $y^k := \text{sor}(y, x_k) = \text{sor}\left(\sum_{j \in S} \beta_j x_j + w, x_k\right)$, for $k = 1, \ldots, p$. Let $m := n/2$. For each $k$, consider the midpoint split of $y^k$ into $y_{[m]}^k$ and $y_{[n] \setminus [m]}^k$ and evaluate the impurity of the

left-half, using empirical variance as impurity:

$$\text{imp}(y_{[m]}^k) := \frac{1}{\binom{m}{2}} \sum_{1 \le i < j \le m} \frac{1}{2}(y_i^k - y_j^k)^2. \tag{1}$$

Let $\text{imp}(y_{[m]}^k)$ be the score of feature $k$, and output the $s$ features with the *smallest* scores (corresponding to maximal reduction in impurity). If the generative model is linear, the choice of the midpoint is justified by our assumption of the uniform distribution for the features $(Z_i)$, and we further show that this simple choice is effective even under a nonlinear model. The choice of the left-half in our analysis is for convenience; a similar analysis applies if we take the impurity to be that of the sum of both halves (or their maximum). DSTUMP is summarized in Algorithm 1. Impurity reduction $\text{imp}(y_{[m]}) - \text{imp}(y_{[m]}^k)$ can be considered a form of nonlinear correlation between $y$ and feature $x_k$. The SIS algorithm is equivalent to replacing this nonlinear correlation with the (absolute) linear correlation $|\frac{1}{n} x_k^T y|$. That is, both procedures assign a score to each feature by considering it against the response separately, ignoring other features. In the uncorrelated (i.e. orthogonal design) setting, this is more or less optimal, and as is the case with SIS, we show that DSTUMP also retains some model selection performance even under correlated designs. In contrast to SIS, we show that DSTUMP also enjoys performance guarantees in non-linear settings.

**The models.** We present our consistency results for models of various complexity. We start with the well-known and extensively studied setting of a linear model with IID design. This basic setup serves as the benchmark for comparison of model selection procedures. As will become clear in the course of the proof, analyzing DSTUMP (or impurity-based feature selection in general) is challenging even in this case, in contrast to linear model based approaches such as SIS or Lasso. Once we have a good understanding of DSTUMP under the baseline model, we extend the analysis to correlated design and nonlinear additive models. The structure of our proof is also most clearly seen in this simple case, as outlined in Section 3.3. We now introduce our general models:

**Model 1** (Sparse linear model with ICA-type design). *A linear regression model $y = X\beta + w$ with ICA-type (random) design $X \in \mathbb{R}^{n \times p}$ has the following properties: (i) $X = \widetilde{X}M$ where $\widetilde{X} \in \mathbb{R}^{n \times p}$ and each row of $\widetilde{X}$ is an independent draw from a (column) vector $Z = (Z_1, \ldots, Z_p)$ with IID entries drawn uniformly from $[0,1]$. (ii) The noise vector $w = (w_1, \ldots, w_n)$ has IID sub-Gaussian entries with variance with variance $\text{var}(w_i) = v_w^2$ and sub-Gaussian norm $\|w_i\|_{\psi_2} \le \sigma_w$, . (iii) The $\beta \in \mathbb{R}^p$ is s-sparse, namely, $\beta_j \ne 0$ for $j \in S = \{1, \ldots, s\}$ and zero otherwise.*

Model 1 serves both the correlated and uncorrelated design cases. Each row of the design matrix $X$ is a draw from the vector $M^T Z$, which has covariance $c\, M^T M$ for some constant $c$. Thus, the choice of $M = I$ leads to an uncorrelated design. The choice of the interval $[0,1]$ for covariates is for convenience; it can be replaced with any other compact interval, in the linear setting, since variance impurity is invariant to a shift. Similarly the choice of the (active) support indices, $S$, is for convenience. For simplicity, we often assume $v_w^2 = \sigma^2$ and $\sigma_w \le C\sigma$ (only $\sigma_w$ would affect the results as examining of our proofs shows).

**Model 2** (Sparse additive model with uncorrelated design). *An additive regression model $y_i = \sum_{j=1}^p f_j(x_{ij}) + w_i$, is one with random design $X = (x_{ij})$ and the noise $(w_i)$ as in Model 1, with $M = I$ (uncorrelated design). We assume $(f_k)$ to be s-sparse, namely, $f_j \ne 0$ for $j \in S = \{1, \ldots, s\}$ and zero otherwise.*

### 3.1 Linear Setting

**Uncorrelated design.** Our baseline result is the following feature selection consistency guarantee for DSTUMP, for the case $M = I$ of Model 1. Throughout, we let $\check{p} := p - s$, and $C, C_1, \ldots, c, c_1, \ldots$ are absolute positive constants which can be different in each occurrence. For any vector $x$, let $|x|_{\min} := \min_i |x_i|$, the minimum absolute value of its entries. The quantity $|\beta_S|_{\min}^2 = \min_{k \in S} \beta_k^2$ appearing in Theorem 1 is a well-known parameter controlling hardness of subset recovery. All our results are stated in terms of constants $\delta, \alpha$ and $\xi$ that are related as:

$$\delta \in (0, 1/8), \quad \alpha = \log(1/(8\delta)), \quad \xi = 1 - (1-\delta)^2. \tag{2}$$

**Theorem 1.** *Assume Model 1 with $M = I$, and (2). The DSTUMP algorithm, which selects the "s" least impure features at the root, succeeds in feature selection, with probability at least $1 - \check{p}^{-c} - 2e^{-\alpha n/2}$ if $\log \check{p}/n \leq C_1$ and*

$$|\beta_S|^2_{\min} \geq \frac{C}{\xi}(\|\beta\|^2_2 + \sigma^2)\sqrt{\frac{\log \check{p}}{n}} \qquad (3)$$

The result can be read by setting, e.g., $\delta = 1/16$ leading to numerical constants for $\alpha$ and $\xi$. The current form allows the flexibility to trade-off the constant ($\alpha$) in the probability bound with the constant ($\xi$) in the gap condition (3). Although Theorem 1 applies to a general $\beta$, it is worthwhile to see its consequence in a special regime of interest where $|\beta_S|^2_{\min} \asymp 1/s$, corresponding to $\|\beta\|_2 \asymp 1$. We get the following immediate corollary:

**Corollary 1.** *Assume $|\beta_S|^2_{\min} \asymp 1/s$, $\sigma^2 \asymp 1$ and $\log \check{p}/n = O(1)$. Then DSTUMP succeeds with high probability if $n \gtrsim s^2 \log \check{p}$.*

The minimax optimal threshold for support recovery in the regime of Corollary 1 is known to be $n \asymp s \log \check{p}$ [11], and achieved by LASSO [12]. Although this result is obtained for Gaussian design, the same argument goes through for our uniform ensemble. Compared to the optimal threshold, using DSTUMP we pay a small factor of $s$ in the sample complexity. However, DSTUMP is not tied to the linear model and as we discuss in Section 3.2, we can generalize the performance of DSTUMP to nonlinear settings.

**Correlated design.** We take the following approach to generalize our result to the correlated case: (1) We show a version of Theorem 1, which holds for an "approximately sparse" parameter $\widetilde{\beta}$ with uncorrelated design. (2) We derive conditions on $M$ such that the correlated case can be turned into the uncorrelated case with approximate sparsity. The following theorem details Step 1:

**Theorem 2.** *Assume Model 1(i)-(ii) with $M = I$, but instead of (iii) let $\beta = \widetilde{\beta}$, a general vector in $\mathbb{R}^p$. Let $S$ be any subset of $[p]$ of cardinality $s$. The DSTUMP algorithm, which selects the "s" least impure features at the root, recovers $S$, with probability at least $1 - \check{p}^{-c} - 2e^{-\alpha n/2}$ if $\log \check{p}/n \leq C_1$ and $\xi|\widetilde{\beta}_S|^2_{\min} - \|\widetilde{\beta}_{S^c}\|^2_\infty > C(\|\widetilde{\beta}\|^2_2 + \sigma^2)\sqrt{(\log \check{p})/n}$.*

The theorem holds for any $\widetilde{\beta}$ and $S$, but the gap condition required is likely to be violated unless $\widetilde{\beta}$ is approximately sparse w.r.t. $S$. Going back to Model 1, we see that setting $\widetilde{\beta} = M\beta$ transforms the model with correlated design $X$, and exact sparsity on $\beta$, to the model with uncorrelated design $\widetilde{X}$, and approximate sparsity on $\widetilde{\beta}$. The following corollary gives sufficient conditions on $M$, so that Theorem 2 is applicable. Recall the usual (vector) $\ell_\infty$ norm, $\|x\|_\infty = \max_i |x_i|$, the matrix $\ell_\infty \to \ell_\infty$ operator norm $\|A\|_\infty = \max_i \sum_j |A_{ij}|$, and the $\ell_2 \to \ell_2$ operator norm $\|A\|_2$.

**Corollary 2.** *Consider a general ICA-type Model 1 with $\beta$ and $M$ satisfying*

$$\|\beta_S\|_\infty \leq \gamma|\beta_S|_{\min}, \quad \|M_{SS} - I\|_\infty \leq \frac{1-\rho}{\gamma}, \quad \|M_{S^cS}\|_\infty \leq \frac{\rho}{\gamma}\sqrt{\xi(1-\kappa)} \qquad (4)$$

*for some $\rho, \kappa \in (0,1]$ and $\gamma \geq 1$. Then, the conclusion of Theorem 1 holds, for DSTUMP applied to input $(y, \widetilde{X})$, under the gap condition (3) with $C/\xi$ replaced with $C\|M_{SS}\|^2_2/(\kappa \xi \rho^2)$.*

Access to decorrelated features, $\widetilde{X}$, is reasonable in cases where one can perform consistent ICA. This assumption is practically plausible, especially in the low-dimensional regimes, though it would be desirable if this assumption can be removed theoretically. Moreover, we note that the response $y$ is based on the correlated features.

In this result, $C\|M_{SS}\|^2_2/(\kappa \xi \rho^2)$ plays the role of a new constant. There is a hard bound on how big $\xi$ can be, which via (4) controls how much correlation between off-support and on-support features are tolerated. For example, taking $\delta = 1/9$, we have $\alpha = \log(9/8) \approx 0.1$, $\xi = 17/81 \approx 0.2$ and $\sqrt{\xi} \approx 0.45$ and this is about as big as it can get (the maximum we can allow is $\approx 0.48$). $\kappa$ can be arbitrarily close to 0, relaxing the assumption (4), at the expense of increasing the constant in the threshold. $\gamma$ controls deviation of $|\beta_j|, j \in S$ from uniform: in case of equal weights on the support, i.e., $|\beta_j| = 1/\sqrt{s}$ for $j \in S$, we have $\gamma = 1$. Theorem 1 for the uncorrelated design is recovered, by taking $\rho = \kappa = 1$.

## 3.2 General Additive Model Setting

To prove results in this more general setting, we need some further regularity conditions on $(f_k)$: Fix some $\delta \in (0,1)$, let $U \sim \mathrm{unif}(0,1)$ and assume the following about the underlying functions $(f_k)$: **(F1)** $\|f_k(\alpha U)\|_{\psi_2}^2 \leq \sigma_{f,k}^2$, $\forall \alpha \in [0,1]$. **(F2)** $\mathrm{var}[f_k(\alpha U)] \leq \mathrm{var}[f_k((1-\delta)U)]$, $\forall \alpha \leq 1 - \delta$.

Next, we define $\sigma_{f,*}^2 := \sum_{k=1}^{p} \sigma_{f,k}^2 = \sum_{k \in S} \sigma_{f,k}^2$ along with the following key *gap* quantities:

$$g_{f,k}(\delta) := \mathrm{var}[f_k(U))] - \mathrm{var}[f_k((1-\delta)U)].$$

**Theorem 3.** *Assume additive Model 2 with (F1) and (F2). Let $\alpha = \log \frac{1}{8\delta}$ for $\delta \in (0, 1/8)$. The* DSTUMP *algorithm, which selects the "s" least impure features at the root, succeeds in model selection, with probability at least $1 - \check{p}^{-c} - 2e^{-\alpha n/2}$ if $\log \check{p}/n \leq C_1$ and*

$$\min_{k \in S} g_{f,k}(\delta) \geq C(\sigma_{f,*}^2 + \sigma^2)\sqrt{\frac{\log \check{p}}{n}} \tag{5}$$

In the supplementary material, we explore in detail the class of functions that satisfy conditions **(F1)** and **(F2)**, as well as the gap condition in (5). **(F1)** is relatively mild and satisfied if $f$ is Lipschitz or bounded. **(F2)** is more stringent and we show that it is satisfied for *convex nondecreasing* and *concave nonincreasing* functions.[2] The gap condition is less restrictive than **(F2)** and is related to the slope of the function near the endpoint, i.e., $x = 1$. Notably, we study one such function that satisfies all of these conditions, i.e., $\exp(\cdot)$ on $[-1, 1]$, in our simulations in Section 4.

## 3.3 Proof of Theorem 1

We provide the high-level proof of Theorem 1. For brevity, the proofs of the lemmas have been omitted and can be found in the supplement, where we in fact prove them for the more general setup of Theorem 3. The analysis boils down to understanding the behavior of $y^k = \mathrm{sor}(y, x_k)$ as defined earlier. Let $\widetilde{y}^k$ be obtained from $y^k$ by random reshuffling of its left half $y_{[m]}^k$ (i.e., rearranging the entries according to a random permutation). This reshuffling has no effect on the impurity, that is, $\mathrm{imp}(\widetilde{y}_{[m]}^k) = \mathrm{imp}(y_{[m]}^k)$, and the reason for it becomes clear when we analyze the case $k \in S$.

**Understanding the distribution of $y^k$.** If $k \notin S$, the ordering according to which we sort $y$ is independent of $y$ (since $x_k$ is independent of $y$), hence the sorted version, before and after reshuffling has the same distribution as $y$. Thus, each entry of $\widetilde{y}^k$ is an IID draw from the same distribution as the pre-sort version:

$$\widetilde{y}_i^k \overset{\mathrm{iid}}{\sim} W_0 := \sum_{j \in S} \beta_j Z_j + w_1, \quad i = 1, \ldots, n. \tag{6}$$

On the other hand, if $k \in S$, then for $i = 1, \ldots, n$

$$y_i^k = \beta_k x_{(i)k} + r_i^k, \quad \text{where} \quad r_i^k \overset{\mathrm{iid}}{\sim} W_k := \sum_{j \in S \setminus \{k\}} \beta_j Z_j + w_1.$$

Here $x_{(i)k}$ is the $i$th order statistic of $x_k$, that is, $x_{(1)k} \leq x_{(2)k} \leq \cdots \leq x_{(n)k}$. Note that the residual terms are still IID since they gather the covariates (and the noise) that are independent of the $k$th one and hence its ordering. Note also that $r_i^k$ is independent of the first term $\beta_k x_{(i)k}$.

Recall that we split at the midpoint and focus on the left split, i.e., we look at $y_{[n/2]}^k = (y_1^k, y_2^k, \ldots, y_{n/2}^k)$, and its reshuffled version $\widetilde{y}_{[n/2]}^k = (\widetilde{y}_1^k, \widetilde{y}_2^k, \ldots, \widetilde{y}_{n/2}^k)$. Intuitively, we would like to claim that the "signal part" of the $\widetilde{y}_{[n/2]}^k$ are approximately IID draws from $\beta_k \mathrm{Unif}(0, 1/2)$. Unfortunately this is not true, in the sense that the distribution cannot be accurately approximated by $\mathrm{Unif}(0, 1 - \delta)$ for any $\delta$ (Lemma 1). However, we show that the distribution can be approximated by an infinite mixture of IID uniforms of reduced range (Lemma 2).

Let $U_{(1)} \leq U_{(2)} \leq \cdots \leq U_{(n)}$ be the order statistics obtained by ordering an IID sample $U_i \sim \mathrm{Unif}(0, 1), i = 1, \ldots, n$. Recall that $m := n/2$ and let $\widetilde{U} := (\widetilde{U}_1, \widetilde{U}_2 \ldots, \widetilde{U}_m)$ be obtained from

$(U_{(1)}, \ldots, U_{(m)})$ by random permutation. Then, $\widetilde{U}$ has an exchangeable distribution. We can write for $k \in S$,

$$\widetilde{y}_i^k = \beta_k \, \widetilde{u}_i^k + \widetilde{r}_i^k, \quad \widetilde{u}^k \sim \widetilde{U}, \quad \text{and} \quad \widetilde{r}_i^k \stackrel{\text{iid}}{\sim} W_k, \; i \in [m]$$

where the $m$-vectors $\widetilde{u}^k = (\widetilde{u}_i^k, i \in [m])$ and $\widetilde{r}^k = (\widetilde{r}_i^k, i \in [m])$ are also independent.

We have the following result regarding the distribution of $\widetilde{U}$:

**Lemma 1.** *The distribution of $\widetilde{U}$ is a mixture of IID* $\mathrm{unif}(0, \gamma)$ *$m$-vectors with mixing variable* $\gamma \sim Beta(m, m + 1)$.

Note that $\mathrm{Beta}(m, m + 1)$ has mean $= m/(2m + 1) = (1 + o(1))/2$ as $m \to \infty$, and variance $= O(m^{-1})$. Thus, Lemma 1 makes our intuition precise in the sense that the distribution of $\widetilde{U}$ is a "range mixture" of IID uniform distributions, with the range concentrating around $1/2$. We now provide a reduced range, finite sample approximation in terms of the total variation distance $d_{\mathrm{TV}}(\widetilde{U}, \widehat{U})$ between the distributions of random vectors $\widetilde{U}$ and $\widehat{U}$.

**Lemma 2.** *Let $\widehat{U}$ be distributed according to a mixture of IID $Unif(0, \widehat{\gamma})$ $m$-vectors with $\widehat{\gamma}$ distributed as a $Beta(m, m + 1)$ truncated to $(0, 1 - \delta)$ for $\delta = e^{-\alpha}/8$ and $\alpha > 0$. With $\widetilde{U}$ as in Lemma 1, we have $d_{TV}(\widetilde{U}, \widehat{U}) \le 2\exp(-\alpha m)$.*

The approximation of the distribution of the $\widetilde{U}$ by a truncated version, $\widehat{U}$, is an essential technique in our proof. As will become clear in the proof of Lemma 3, we will need to condition on the mixing variable $\widetilde{U}$, or its truncated approximation $\widehat{U}$, to allow for the use of concentration inequalities for independent variables. The resulting bounds should be devoid of randomness so that by taking expectation, we can get similar bounds for the exchangeable case. The truncation allows us to maintain a positive gap in impurities (between on and off support features) throughout this process. We expect the loss due to truncation to be minimal, only impacting the constants.

For $k \in S$, let $\widehat{u}^k = (\widehat{u}_i^k, i \in [m])$ be drawn from the distribution of $\widehat{U}$ described in Lemma 2, independently of anything else in the model, and let $\widehat{\gamma}^k$ be its corresponding mixing variable, which has a Beta distribution truncated to $(0, 1 - \delta)$. Let us define $\widehat{y}_i^k = \beta_k \, \widehat{u}_i^k + \widetilde{r}_i^k, \; i \in [m]$ where $\widetilde{r}^k = (\widetilde{r}_i^k)$ is as before. This construction provides a simple coupling between $\widetilde{y}_{[m]}^k$ and $\widehat{y}_{[m]}^k$ giving the same bound on the their total variation distance. Hence, we can safely work with $\widehat{y}_{[m]}^k$ instead of $\widetilde{y}_{[m]}^k$, and pay a price of at most $2\exp(-\alpha m)$ in probability bounds. To simplify discussion, let $\widehat{y}_i^k = \widetilde{y}_i^k$ for $k \notin S$.

**Concentration of empirical impurity.** We will focus on $\widehat{y}_{[m]}^k$ due the discussion above. We would like to control $\mathrm{imp}(\widehat{y}_{[m]}^k)$, the empirical variance impurity of $\widehat{y}_{[m]}^k$ which is defined as in (1) with $y_{[m]}^k$ replaced with $\widehat{y}_{[m]}^k$. The idea is to analyze $\mathbb{E}[\mathrm{imp}(\widehat{y}_{[m]}^k)]$, or proper bounds on it, and then show that $\mathrm{imp}(\widehat{y}_{[m]}^k)$ concentrates around its mean. Let us consider the concentration first. (1) is a U-statistic of order 2 with kernel $h(u, v) = \frac{1}{2}(u - v)^2$. The classical Hoeffding inequality guarantees concentration if $h$ is uniformly bounded and the underlying variables are IID. Instead, we use a version of Hanson–Wright concentration inequality derived in [10], which allows us to derive a concentration bound for the empirical variance, for general sub-Gaussian vectors, avoiding the boundedness assumption:

**Corollary 3.** *Let $w = (w_1, \ldots, w_m) \in \mathbb{R}^m$ be a random vector with independent components $w_i$ which satisfy $\mathbb{E} w_i = \mu$ and $\|w_i - \mu\|_{\psi_2} \le K$. Let $\mathrm{imp}(w) := \binom{m}{2}^{-1} \sum_{1 \le i < j \le m} (w_i - w_j)^2$ be the empirical variance of $w$. Then, for $u \ge 0$,*

$$\mathbb{P}\Big(\big|\mathrm{imp}(w) - \mathbb{E}\,\mathrm{imp}(w)\big| > K^2 u\Big) \le 2\exp\big\{-c\,(m - 1)\min(u, u^2)\big\}. \tag{7}$$

We can immediately apply this result when $k \notin S$. However, for $k \in S$, a more careful application is needed since we can only guarantee an exchangeable distribution for $\widehat{y}_{[m]}^k$ in this case. The following lemma summarizes the conclusions:

**Lemma 3.** *Let $\widehat{I}_{m,k} = \mathrm{imp}(\widehat{y}_{[m]}^{k})$ and recall that $\delta$ was introduced in the definition of $\widehat{y}_{i}^{k}$. Let $\kappa_1^2 := \frac{1}{12}$ be the variance of $Unif(0,1)$. Recall that $\check{p} := p - s$. Let $L = \|\beta\|_2$. There exist absolute constants $C_1, C_2, c$ such that if $\log \check{p}/m \le C_1$, then with probability at least $1 - \check{p}^{-c}$,*

$$\widehat{I}_{m,k} \le I_k^1 + \varepsilon_m, \ \forall k \in S, \quad and, \quad \widehat{I}_{m,k} \ge I^0 - \varepsilon_m, \ \forall k \notin S$$

*where, letting $\xi := 1 - (1 - \delta)^2$,*

$$I_k^1 := \kappa_1^2(-\xi\beta_k^2 + L^2) + \sigma^2, \quad I^0 := \kappa_1^2 L^2 + \sigma^2, \ and \ \varepsilon_m := C_2(L^2 + \sigma^2)\sqrt{\log \check{p}/m}.$$

The key outcome of Lemma 3 is that, on average, there is a positive gap $I^0 - I_k^1 = \kappa_1^2\xi\beta_k^2$ in impurities between a feature on the support and those off of it, and that due to concentration, the fluctuations in impurities will be less than this gap for large $m$. Combined with Lemma 2, we can transfer the results to $\widetilde{I}_{m,k} := \mathrm{imp}(\widetilde{y}_{[m]}^{k})$.

**Corollary 4.** *The conclusion of Lemma 3 holds for $\widetilde{I}_{m,k}$ in place of $\widehat{I}_{m,k}$, with probability at least $1 - \check{p}^{-c} - 2e^{-\alpha m}$ for $\alpha = \log \frac{1}{8\delta}$.*

Note that for $\delta < 1/8$, the bound holds with high probability. Thus, as long as $I^0 - I_k^1 > 2\varepsilon_m$, the selection algorithm correctly favors the $k$th feature in $S$, over the inactive ones (recall that lower impurity is better). We have our main result after substituting $n/2$ for $m$.

## 4 Simulations

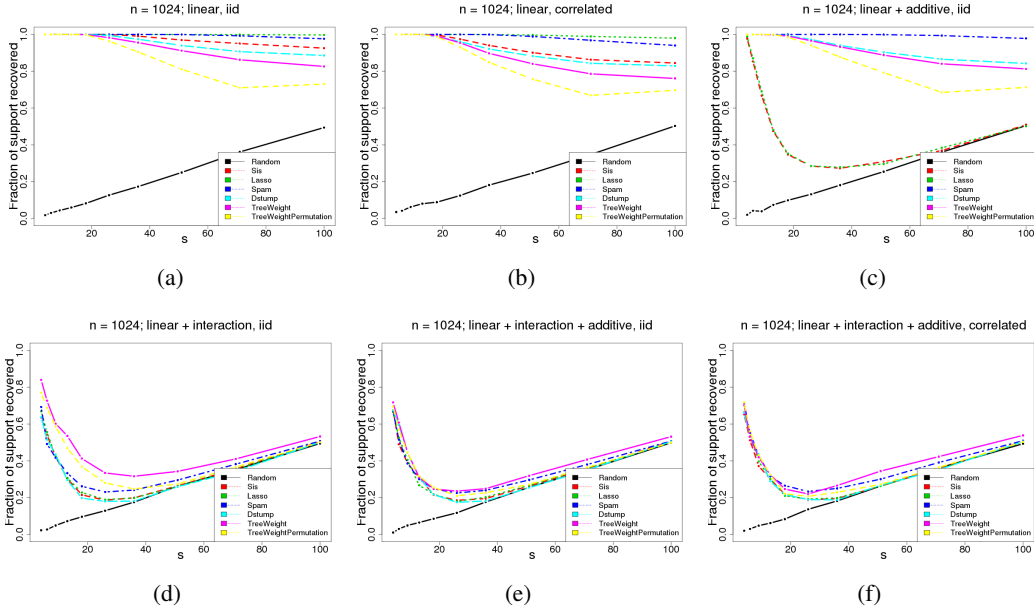

(a)           (b)           (c)

(d)           (e)           (f)

Figure 1: Support recovery performance in a linear regression model augmented with possible nonlinearities for $n = 1024$. (a) Linear case with uncorrelated design. (b) Linear case with correlated design. (c) Nonlinear additive model with exponentials of covariates and uncorrelated design. (d) Nonlinear model with interaction terms and uncorrelated design. (e) Nonlinear additive model with exponentials of covariates, interaction terms, and uncorrelated design. (f) Nonlinear additive model with exponentials of covariates, interaction terms, and correlated design.

In order to corroborate the theoretical analysis, we next present various simulation results. We consider the following model: $y = X\beta + f(X_S) + w$, where $f(X_S)$ is a potential nonlinearity, and $S$ is the true support of $\beta$. We generate the training data as $X = \widetilde{X}M$ where $\widetilde{X} \in \mathbb{R}^{n \times p}$ is a random matrix with IID $Unif(-1, 1)$ entries, and $M \in \mathbb{R}^{p \times p}$ is an upper-triangular matrix that determines whether the design is IID or correlated. In the IID case we set $M = I$. To achieve a correlated design

we randomly assign values from $\{0, -\rho, +\rho\}$ to the upper triangular cells of $M$, with probabilities $(1 - 2\alpha, \alpha, \alpha)$. We observed qualitatively similar results for various values of $\rho$ and $\alpha$ and here we present results with $\alpha = 0.04$, and $\rho = 0.1$. The noise is generated as $w \sim N(0, \sigma^2 I_n)$. We fix $p = 200$, $\sigma = 0.1$, and let $\beta_i = \pm 1/\sqrt{s}$ over its support $i \in S$, where $|S| = s$. That is, only $s$ of the $p = 200$ variables are predictive of the response. The nonlinearity, $f$, optionally contains additive terms in the form of exponentials of on-support covariates. It can also contain interaction terms across on-support covariates, i.e., terms of the form $\frac{2}{\sqrt{s}} x_i x_j$ for some randomly selected pairs of $i, j \in S$. Notably, the choice of $f$ is unknown to the variable selection methods. We vary $s \in [5, 100]$ and note that $\|\beta\|_2 = 1$ remains fixed.

The plots in Figure 1 show the fraction of the true support recovered[3] as a function of $s$, for various methods under different modeling setups: $f = 0$ (*linear*), $f = 2\exp(\cdot)$ (*additive*), $f =$ interaction (*interactions*), and $f =$ interaction $+ 2\exp(\cdot)$ (*interactions+additive*) with IID or correlated designs. Each data point is an average over 100 trials (see supplementary material for results with 95% confidence intervals). In addition to DSTUMP, we evaluate TREEWEIGHT, SPAM, LASSO, SIS and random guessing for comparison. SIS refers to picking the indices of the top $s$ largest values of $X^T y$ in absolute value. When $X$ is orthogonal and the generative model is linear, this approach is optimal, and we use it as a surrogate for the optimal approach in our *nearly orthogonal* setup (i.e., the IID linear case), due to its lack of any tuning parameters. Random guessing is used as a benchmark, and as expected, on average recovers the fraction $s/p = s/200$ of the support.

The plots show that, in the linear setting, the performance of DSTUMP is comparable to, and only slightly worse than, that of SIS or Lasso which are considered optimal in this case. Figure 1(b) shows that under mildly correlated design the gap between DSTUMP and LASSO widens. In this case, SIS loses its optimality and performs at the same level as DSTUMP. This matches our intuition as both SIS and DSTUMP are both greedy methods that consider covariates independently.

DSTUMP is more robust to nonlinearities, as characterized theoretically in Theorem 3 and evidenced in Figure 1(c). In contrast, in the presence of exponential nonlinearities, SIS and Lasso are effective in the very sparse regime of $s \ll p$, but quickly approach random guessing as $s$ grows. In the presence of interaction terms, TREEWEIGHT and to a lesser extent SPAM outperform all other methods, as shown in Figure 1(d), 1(e), and 1(f). We also note that the permutation-based importance method [1], denoted by TREEWEIGHTPERMUTATION in the plots in Figure 1, performs substantially worse than TREEWEIGHT across the various modelling settings.

Overall, these simulations illustrate the promise of multi-level tree-based methods like TREEWEIGHT under more challenging and realistic modeling settings. Future work involves generalizing our theoretical analyses to extend to these more complex multi-level tree-based approaches.

## 5  Discussion

We presented a simple model selection algorithm for decision trees, which we called DSTUMP, and analyzed its finite-sample performance in a variety of settings, including the high-dimensional, nonlinear additive model setting. Our theoretical and experimental results show that even a simple tree-based algorithm that selects at the root can achieve high dimensional selection consistency.

We hope these results pave the way for the finite-sample analysis of more refined tree-based model selection procedures. Inspired by the empirical success of TREEWEIGHT in nonlinear settings, we are actively looking at extensions of DSTUMP to a multi-stage algorithm capable of handling interactions with high-dimensional guarantees.

Moreover, while we mainly focused on the regression problem, our proof technique based on concentration of impurity reductions is quite general. We expect analogous results to hold, for example for classification. However, aspects of the proof would be different, since impurity measures used for classification are different than those of regression. One major hurdle involves deriving concentration inequalities for the empirical versions of these measures, which are currently unavailable, and would be of independent interest.

## Footnotes

*Now at Google

[2]We also observe that this condition holds for functions beyond these two categories.

[3]In the supplementary material we report analogous results using a more stringent performance metric, namely the probability of exact support recovery. The results are qualitatively similar.

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
