[Supplementary Material · dstump_analysis_v6.pdf]

# Variable Importance using Decision Trees, Supplementary Material for NIPS 2017 paper

S. Jalil Kazemitabar[*], Arash A. Amini[*], Adam Bloniarz[†], Ameet Talwalkar[‡]

We present a complete presentation of the theoretical results presented in the main text. We provide detailed analysis of the DSTUMP algorithm in the context of a general additive regression model with uncorrelated design. We derive the results for the linear case as special case of the general theory. Our analysis is high-dimensional and non-asymptotic, and to our knowledge the first such analysis of feature selection properties of decision trees. We show that even in the high-dimensional setting where the number of features $p$ is much larger that the sample size $n$, feature importance scores based on impurity reduction contain enough information for consistent model selection. Additionally, we provide simulation experiments to supplement the results in the main text.

## 1   Setup

Consider an additive regression model $y_i = \sum_{j=1}^{p} f_j(x_{ij}) + w_i$ with random design $X = (x_{ij}) \in \mathbb{R}^{n \times p}$. Here each column of $X$ represents a covariate or feature. Let us denote generic covariates as $Z_1, \ldots, Z_p$ and assume them to be i.i.d. uniformly distributed on $[0, 1]$. Each row of $X$ is taken to be an independent draw from $(Z_1, \ldots, Z_p)$. We assume $(f_k)$ to be $s$-sparse, namely, $f_j \neq 0$ for $j \in S = \{1, \ldots, s\}$ and zero otherwise. Let $w = (w_1, \ldots, w_n)$ and assume the entries to be IID draws from a sub-Gaussian distribution with variance $\text{var}(w_i) = v_w^2$ and sub-Gaussian norm $\|w_i\|_{\psi_2} \leq \sigma_w^2$, for all $i \in [n]$. Recall that the sub-Gaussian norm is defined as (see [Ver12]):

$$\|w_i\|_{\psi_2} := \sup_{k \geq 1} k^{-1/2} (\mathbb{E}|w_i|^k)^{1/k}.$$

Fix some $\delta \in (0, 1)$, let $U \sim \text{unif}(0, 1)$ and assume the following about the underlying functions $(f_k)$:

**F1**  $\|f_k(\alpha U)\|_{\psi_2}^2 \leq \sigma_{f,k}^2, \quad \forall \alpha \in [0, 1].$

**F2**  $\text{var}[f_k(\alpha U)] \leq \text{var}[f_k((1 - \delta)U)], \ \forall \alpha \leq 1 - \delta.$

Let us define $v_{f,k}^2 := \text{var}[f_k(U)]$ and

$$\sigma_{f,*}^2 := \sum_{k=1}^{p} \sigma_{f,k}^2 = \sum_{k \in S} \sigma_{f,k}^2, \quad v_{f,*}^2 := \sum_{k=1}^{p} v_{f,k}^2 = \sum_{k \in S} v_{f,k}^2 \tag{1}$$

---

[*]UCLA; [†]UC Berkeley; [‡]CMU

Note that we take $\sigma_{f,k}^2 = v_{f,k}^2 = 0$ for $k \notin S$. We also define the following key *gap* quantities,

$$g_{f,k}(\delta) := \mathrm{var}[f_k(U))] - \mathrm{var}[f_k((1-\delta)U)]. \tag{2}$$

Section 3 explores the class of functions satisfying (F1) and (F2) and have positive gap: $g_{f,k}(\delta) > 0$.

Throughout, $C, C_1, \ldots, c, c_1, \ldots$ are absolute positive constants which can be different in each occurrence.

**Facts about sub-Gaussian vectors [Ver12].** Recall that if $\{X_i\}$ are independent zero-mean sub-Gaussian variables then so is $\sum_i X_i$ and we have

**SG-1** $\|\sum_i X_i\|_{\psi_2}^2 \le C \sum_i \|X_i\|_{\psi_2}^2$.

Centering inflates the sub-Gaussian norm by at most a factor of 2, i.e.,

**SG-2** $\|X - \mathbb{E}X\|_{\psi_2} \le 2\|X\|_{\psi_2}$.

Examples of sub-Gaussian random variables included bounded and Gaussian variables:

**SG-3** If $|X| \le K$ a.s., then $X$ is sub-Gaussian with norm $\|X\|_{\psi_2} \le K$.

**SG-4** $X \sim N(0, \sigma^2)$ is sub-Gaussian with $\|X_i\|_{\psi_2} \le C\sigma$ for absolute constant $C$.

**Notation.** In order describe DSTUMP more precisely, let us introduce some notation. We write $[p] := \{1, \ldots, p\}$. For an ordered index set $\mathcal{I} = (i_1, i_2, \ldots, i_r)$, we set $y_{\mathcal{I}} = (y_{i_1}, y_{i_2} \ldots, y_{i_r})$. The order of elements of $y_{\mathcal{I}}$ matter in this case. For an unordered index set $S = \{i_1, i_2, \ldots, i_r\}$, we first turn $S$ into an ordered set, with elements in increasing order, and then form $y_S$. We write $x_j = (x_{1j}, x_{2j}, \ldots, x_{nj}) \in \mathbb{R}^n$ for the vector collecting values of the $j$th feature, i.e., the $j$th column of $X$. Let $\mathcal{I}(x_j) := (i_1, i_2 \ldots, i_n)$ be an ordering of $[n]$ such that $x_{i_1 j} \le x_{i_2 j} \le \cdots \le x_{i_n j}$ and let $\mathrm{sor}(y, x_j) := y_{\mathcal{I}(x_j)} \in \mathbb{R}^n$; this is an operator that sorts $y$ relative to $x_j$.

DSTUMP proceeds as follows: Evaluate

$$y^k := \mathrm{sor}(y, x_k) = \mathrm{sor}\left(\sum_{j \in S} f_j(x_j) + w, x_k\right) \tag{3}$$

for $k = 1, \ldots, p$. Let $m := n/2$. For each $k$, consider the midpoint split of $y^k$ into $y_{[m]}^k$ and $y_{[n]\setminus[m]}^k$ and evaluate the impurity of the left-half, using empirical variance as impurity:

$$\mathrm{imp}(y_{[m]}^k) := \frac{1}{\binom{m}{2}} \sum_{1 \le i < j \le m} \frac{1}{2}(y_i^k - y_j^k)^2 \tag{4}$$

Let $\mathrm{imp}(y_{[m]}^k)$ be the score of feature $k$, and output the $s$ features with the *least* scores (corresponding to maximal reduction in impurity). The choice of the midpoint is justified by our assumption of the uniform distribution for the features $(Z_i)$. The choice of the left-half is for convenience; a similar analysis applies if we take the impurity to be that of the sum of both halves (or their maximum).

# 2 Analysis of DStump

The problem boils down to understanding the behavior of $y^k$. Let $\widetilde{y}^k$ be obtained from $y^k$ by random reshuffling of its left half $y^k_{[m]}$, i.e., rearranging the entries according to a random permutation. This reshuffling has no effect on the impurity, that is, $\text{imp}(\widetilde{y}^k_{[m]}) = \text{imp}(y^k_{[m]})$ and is done so that $\widetilde{y}^k_{[m]}$ has an exchangeable distribution (cf. Lemma 1). Throughout, we assume $m \geq 2$.

### 2.0.1  Understanding the distribution of $y^k$

If $k \notin S$, the ordering according to which we sort $y$ is independent of $y$ (since $x_k$ is independent of $y$), hence the sorted version, before and after reshuffling has the same distribution as $y$. Thus, each entry of $\widetilde{y}^k$ is an IID draw from the same distribution as the pre-sort version:

$$\widetilde{y}^k_i \overset{\text{iid}}{\sim} W_0 := \sum_{j \in S} f_j(Z_j) + w_1, \quad i = 1, \dots, n. \tag{5}$$

On the other hand, if $k \in S$, then for $i = 1, \dots, n$

$$y^k_i = f_k(x_{(i)k}) + r^k_i, \qquad r^k_i \overset{\text{iid}}{\sim} W_k := \sum_{j \in S \setminus \{k\}} f_j(Z_j) + w_1. \tag{6}$$

Here $x_{(i)k}$ is the $i$th order statistic of $x_k$, that is, $x_{(1)k} \leq x_{(2)k} \leq \cdots \leq x_{(n)k}$. Note that the residual terms are still IID since they gather the covariates (and the noise) that are independent of the $k$th one and hence its ordering. Note also that $r^k_i$ is independent of the first term $f_k(x_{(i)k})$.

Using the convention $f_k \equiv 0$ for $k \notin S$, we can combine (5) and (6) into a single equation that works for all $k \in [p]$, (and $i \in [m]$),

$$y^k_i = f_k(x_{(i)k}) + r^k_i, \qquad r^k_i \overset{\text{iid}}{\sim} W_k := \sum_{j \neq k} f_j(Z_j) + w_1. \tag{7}$$

for all $k \in [p]$. Note that $W_k = W_0 = \sum_{j \in S} f_j(Z_i)$ for all $k \notin S$.

Recall that we split at the midpoint and focus on the left split, i.e., we look at $y^k_{[n/2]} = (y^k_1, y^k_2, \dots, y^k_{n/2})$, and its reshuffled version $\widetilde{y}^k_{[n/2]} = (\widetilde{y}^k_1, \widetilde{y}^k_2, \dots, \widetilde{y}^k_{n/2})$. Intuitively, we would like to claim that the signal part of $\widetilde{y}^k_{[n/2]}$ has entries that are approximately IID draws from $f_k(\frac{1}{2}U)$ where $U \sim \text{unif}(0, 1)$. Unfortunately this is not true, in the sense that the distribution cannot be accurately approximated by $\text{Unif}(0, 1 - \delta)$ for any $\delta$ (Lemma 1). However, we show that the distribution can be approximated by an infinite mixture of IID uniforms of reduced range (Lemma 2). Proofs of Lemma 1 and 2 appear in Appendix 5.

### 2.0.2  Truncating the range

Let $U_{(1)} \leq U_{(2)} \leq \cdots \leq U_{(n)}$ be the order statistics obtained by ordering an IID sample $U_i \sim \text{unif}(0, 1), i = 1, \dots, n$. Recall that $m := n/2$ and let $\widetilde{U} := (\widetilde{U}_1, \widetilde{U}_2 \dots, \widetilde{U}_m)$ be obtained from $(U_{(1)}, \dots, U_{(m)})$ by random permutation. Then, $\widetilde{U}$ has an exchangeable distribution. Using (7), we can write for $k \in [p]$,

$$\widetilde{y}^k_i = f_k(\widetilde{u}^k_i) + \widetilde{r}^k_i, \quad \widetilde{u}^k \sim \widetilde{U}, \quad \widetilde{r}^k_i \overset{\text{iid}}{\sim} W_k, \quad i = 1, \dots, m \tag{8}$$

where the $m$-vectors $\widetilde{u}^k = (\widetilde{u}_i^k, i \in [m])$ and $\widetilde{r}^k = (\widetilde{r}_i^k, i \in [m])$ are also independent. We have the following result regarding the distribution of $\widetilde{U}$:

**Lemma 1.** *The density of $\widetilde{U}$ (w.r.t Lebesgue measure on $\mathbb{R}^m$) is given by*

$$(u_1, \ldots, u_m) \mapsto \binom{2m}{m} (1 - \max\{u_1, \ldots, u_m\})^m \tag{9}$$

*over $[0,1]^m$. Furthermore the distribution of $\widetilde{U}$ is a mixture of IID $\mathrm{unif}(0, \gamma)$ $m$-vectors with mixing variable $\gamma \sim Beta(m, m+1)$.*

Note that $\mathrm{Beta}(m, m+1)$ has mean $= m/(2m+1) = \frac{1}{2}(1 + o(1))$ as $m \to \infty$, and variance $= O(m^{-1})$. Thus, Lemma 1 makes our intuition precise in the sense that the distribution of $\widetilde{U}$ is a "range mixture" of IID uniform distributions, with the range concentrating around $1/2$. We now provide a reduced range, finite sample approximation in terms of the total variation distance $d_{\mathrm{TV}}(\widetilde{U}, \widehat{U})$ between the distributions of random vectors $\widetilde{U}$ and $\widehat{U}$.

**Lemma 2.** *Let $\widehat{U}$ be distributed according to a mixture of IID $Unif(0, \widehat{\gamma})$ $m$-vectors with $\widehat{\gamma}$ distributed as a $Beta(m, m+1)$ truncated to $(0, 1 - \delta)$ for $\delta = e^{-\alpha}/8$ and $\alpha > 0$. With $\widetilde{U}$ as in Lemma 1, we have $d_{TV}(\widetilde{U}, \widehat{U}) \leq 2 \exp(-\alpha m)$.*

For $k \in [p]$, let $\widehat{u}^k = (\widehat{u}_i^k, i \in [m])$ be drawn from the distribution of $\widehat{U}$ described in Lemma 2, independently of anything else in the model, and let $\widehat{\gamma}^k$ be its corresponding mixing variable, which has a Beta distribution truncated to $(0, 1 - \delta)$. Let us define

$$\widehat{y}_i^k = f_k(\widehat{u}_i^k) + \widetilde{r}_i^k, \ i \in [m] \tag{10}$$

where $\widetilde{r}^k = (\widetilde{r}_i^k)$ is as before. This construction provides a simple coupling between $\widetilde{y}_{[m]}^k$ and $\widehat{y}_{[m]}^k$ giving the same bound on the their total variation distance as in Lemma 2. Hence, we can safely work with $\widehat{y}_{[m]}^k$ instead of $\widetilde{y}_{[m]}^k$, and pay a price of at most $2\exp(-\alpha m)$ in probability bounds.

### 2.0.3 Concentration of the empirical impurity

We will focus on $\widehat{y}_{[m]}^k$ due the discussion above. We would like to control $\mathrm{imp}(\widehat{y}_{[m]}^k)$, the empirical variance impurity of $\widehat{y}_{[m]}^k$ which is defined as in (4) with $y_{[m]}^k$ replaced with $\widehat{y}_{[m]}^k$. The idea is to analyze $\mathbb{E}[\mathrm{imp}(\widehat{y}_{[m]}^k)]$, or proper bounds on it, and then show that $\mathrm{imp}(\widehat{y}_{[m]}^k)$ concentrates around its mean. Let us consider the concentration first. (4) is a U-statistic of order 2 with kernel $h(u, v) = \frac{1}{2}(u - v)^2$. The classical Hoeffding inequality guarantees concentration if $h$ is uniformly bounded and the underlying variables are IID. Instead, we use a version of Hanson–Wright concentration inequality derived in [RV13]:

**Theorem 1** (Hanson–Wright, Rudelson–Vershynin)**.** *Let $v = (v_1, \ldots, v_m) \in \mathbb{R}^m$ be a random vector with independent components $w_i$, each of which satisfies $\mathbb{E}v_i = 0$ and $\|v_i\|_{\psi_2} \leq K$. Let $A$ be an $m \times m$ matrix. Then, for any $t \geq 0$,*

$$\mathbb{P}\Big(\big|v^T A v - \mathbb{E}v^T A v\big| > t\Big) \leq 2 \exp\left[-c \min\left(\frac{t^2}{K^4\|A\|_F^2}, \frac{t}{K^2\|A\|_{op}}\right)\right]. \tag{11}$$

As a consequence we can get a concentration bound for the empirical variance, for general sub-Gaussian vectors, avoiding the boundedness assumption:

**Corollary 1.** *Let $v = (v_1, \ldots, v_m) \in \mathbb{R}^m$ be a random vector with independent components $v_i$ which satisfy $\mathbb{E} v_i = \mu$ and $\|v_i - \mu\|_{\psi_2} \leq K$. Let $\mathrm{imp}(v) := \binom{m}{2}^{-1} \sum_{1 \leq i < j \leq m} (v_i - v_j)^2$ be the empirical variance of $v$. Then, for $u \geq 0$,*

$$\mathbb{P}\Big( \big| \mathrm{imp}(v) - \mathbb{E}\,\mathrm{imp}(v) \big| > K^2 u \Big) \leq 2 \exp \big\{ -c\,(m-1) \min(u, u^2) \big\}. \tag{12}$$

*Proof.* Since empirical variance is invariant to a constant shift, without loss of generality we can consider only the case $\mu = 0$. We have $\mathrm{imp}(v) = \binom{m}{2}^{-1} \big[ \frac{1}{2}(m-1) \sum_i v_i^2 - \sum_{i<j} v_i v_j \big] = v^T A v$, where $A = \binom{m}{2}^{-1} (\frac{1}{2} m I - \frac{1}{2} \mathbf{1}\mathbf{1}^T)$. Letting $\widetilde{A} = \frac{2}{m} \binom{m}{2} A = I - \frac{1}{m} \mathbf{1}\mathbf{1}^T$, we note that $\widetilde{A}$ has one eigenvalue equal 0, and $m-1$ eigenvalues equal 1. It is positive semidefinite matrix with same singular values. Hence, $\|\widetilde{A}\|_{op} = 1$ and $\|\widetilde{A}\|_F = \sqrt{m-1}$. Applying Theorem 1 to $\widetilde{A}$ and setting $t = (m-1) K^2 u$ gives the result after some algebra. $\qquad\square$

We can immediately apply this result when $k \notin S$. However, for $k \in S$, a more careful application is needed since we can only guarantee an exchangeable distribution for $\widehat{y}_{[m]}^k$ in this case. The following lemma summarizes the conclusions:

Let $\widehat{I}_{m,k} = \mathrm{imp}(\widehat{y}_{[m]}^k)$ and recall that $\delta$ was introduced in the definition of $\widehat{y}_i^k$ in (10). We also recall the definitions of variances and sub-Gaussian norms from (1), and the gaps $g_{f,k}(\delta)$ from (2).

**Lemma 3.** *There exist absolute constants $C_1, C_2, c$ such that if $\log p / m \leq C_1$, then with probability at least $1 - p^{-c}$,*

$$\widehat{I}_{m,k} \in \big( I_k^0 - \varepsilon_m, I_k^1 + \varepsilon_m \big), \; \forall k \in [p] \tag{13}$$

*where*

$$I_k^1 = -g_{f,k}(\delta) + v_{f,*}^2 + v_w^2, \quad I_k^0 := -v_{f,k}^2 + v_{f,*}^2 + v_w^2, \quad \varepsilon_m := C_2(\sigma_{f,*}^2 + \sigma_w^2)\sqrt{\log p / m}. \tag{14}$$

Note that we can replace $p$ with $\check{p} := p - s$ in the bounds at the expense of constants since $\check{p} \leq p \leq 2\check{p}$ assuming $s \leq p/2$.

*Proof of Lemma 3.* Fix $k \in [p]$, and recall (10) and the underlying mixing variable $\widehat{\gamma}^k$. Then, conditioned on $\widehat{\gamma}^k$, $\widehat{y}_i^k, i \in [m]$ is an IID sequence by the definition:

$$\widehat{y}_i^k \mid \widehat{\gamma}^k \overset{\mathrm{iid}}{\sim} f_k(V) + W_k, \quad i = 1, \ldots, m$$

where $V \sim \mathrm{Unif}(0, \widehat{\gamma}^k)$, independent of $W_k = \sum_{j \neq k} f_j(Z_j) + w_1$. We also write $V = \widehat{\gamma}^k U$ for $U \sim \mathrm{unif}(0,1)$. Note that $V$ is defined conditionally on $\widehat{\gamma}^k$, and whenever we are working with $V$ and quantities involving it, we have the implicit conditioning on $\widehat{\gamma}^k$ (without writing it down).

To simplify notation, let $R_k := f_k(V) + W_k$. Using (SG-1), we have

$$\begin{aligned}
\|R_k\|_{\psi_2}^2 := \|f_k(V) + W_k\|_{\psi_2}^2 &\leq C\Big[ \|f_k(V)\|_{\psi_2}^2 + \sum_{j \neq k} \|f_j(Z_j)\|_{\psi_2}^2 + \|w_1\|_{\psi_2}^2 \Big] \\
&\leq C\Big[ \sum_j \sigma_{f,j}^2 + \sigma_w^2 \Big] = C(\sigma_{f,*}^2 + \sigma_w^2)
\end{aligned} \tag{15}$$

In the second inequality, we have used Assumption (F1) to write $\|f_k(V)\|_{\psi_2}^2 = \|f_k(\widehat{\gamma}^k U)\|_{\psi_2}^2 \leq \sigma_{f,k}^2$, since $\widehat{\gamma}^k \leq 1 - \delta < 1$ by definition (and we are working conditioned on $\widehat{\gamma}^k$, making it a constant.) In addition, $\|w_1\|_{\psi_2}^2 \leq \sigma_w^2$ and by Assumption (F1), $\|f_j(Z_j)\|_{\psi_2} \leq \sigma_{f,j}^2$, which combined gives (15) (recalling that $\sigma_{f,*}^2 = \sum_j \sigma_{f,j}^2$).

Note that conditioned on $\widehat{\gamma}^k$, $\widehat{I}_{m,k}$ is of the form $\mathrm{imp}(w)$ for $w_i \overset{\mathrm{iid}}{\sim} R_k, i = 1, \ldots, m$, and we have $\|R_k - \mathbb{E}R_k\|_{\psi_2}^2 \leq 4C(\sigma_{f,*}^2 + \sigma_w^2)$ by (15) and (SG-2). Hence, the conditions of Corollary 1 are met and we obtain, for $k \in [p]$,

$$\mathbb{P}\Big(\big|\widehat{I}_{m,k} - \mathbb{E}[\widehat{I}_{m,k}|\widehat{\gamma}^k]\big| > 4C(\sigma_{f,*}^2 + \sigma_w^2)u \,\Big|\, \widehat{\gamma}^k\Big) \leq 2\exp\big[-c(m-1)\min\{u, u^2\}\big].$$

Moreover, we can bound the expectation as follows:

$$\begin{aligned}
\mathbb{E}[\widehat{I}_{m,k}|\widehat{\gamma}^k] = \mathrm{var}(f_k(V) + W_k) &= \mathrm{var}(f_k(V)) + \mathrm{var}(W_k) \\
&= \mathrm{var}(f_k(V)) + \sum_{j \neq k} \mathrm{var}[f_j(Z_j)] + v_w^2 \\
&\leq \mathrm{var}[f_k((1-\delta)U)] + \sum_{j \neq k} v_{f,j}^2 + v_w^2 \\
&= -g_{f,k}(\delta) + \sum_j v_{f,j}^2 + v_w^2
\end{aligned}$$

where the inequality follows since $\mathrm{var}(f_k(V)) = \mathrm{var}[f_k(\widehat{\gamma}^k U)] \leq \mathrm{var}[f_k((1-\delta)U)]$, using $\widehat{\gamma}^k \leq 1-\delta$ and Assumption (F2). The last equality follows by adding and subtracting $\mathrm{var}[f_k(U)]$ and using the definition of $g_{f,j}(\delta) := \mathrm{var}[f_k(U)] - \mathrm{var}[f_k((1-\delta)U)]$. Thus, recalling the definition of $v_{f,*}$ from (1),

$$\mathbb{E}[\widehat{I}_{m,k}|\widehat{\gamma}^k] \leq -g_{f,k}(\delta) + v_{f,*}^2 + v_w^2 = I_k^1.$$

Similarly $\mathbb{E}[\widehat{I}_{m,k}|\widehat{\gamma}^k] \geq \sum_{j \neq k} v_{f,j}^2 + \sigma_w^2 = -v_{f,k}^2 + v_{f,*}^2 + v_w^2 = I_k^0$. It follows that

$$\big\{\widehat{I}_{m,k} \notin (I_k^0 - t, I_k^1 + t)\big\} \subset \big\{\widehat{I}_{m,k} \notin (\mathbb{E}[\widehat{I}_{m,k}|\widehat{\gamma}^k] - t, \mathbb{E}[\widehat{I}_{m,k}|\widehat{\gamma}^k] + t)\big\}$$

for any $t$. Letting $\varepsilon_m(u) := 4C(\sigma_{f,*}^2 + \sigma_w^2)u$, we obtain

$$\begin{aligned}
\mathbb{P}\Big(\widehat{I}_{m,k} \notin \big(I_k^0 - \varepsilon_m(u), I_k^1 + \varepsilon_m(u)\big)\,\Big|\, \widehat{\gamma}^k\Big) &\leq \mathbb{P}\Big(\big|\widehat{I}_{m,k} - \mathbb{E}[\widehat{I}_{m,k}|\widehat{\gamma}^k]\big| > \varepsilon_m(u)\,\Big|\,\widehat{\gamma}^k\Big) \\
&\leq 2\exp\big[-c(m-1)\min\{u,u^2\}\big].
\end{aligned}$$

Notably, $I_k^0, I_k^1, C, \sigma_{f,*}^2, \sigma_w^2, u$, and $m$ are all constant w.r.t. to $\widehat{\gamma}^k$, so taking the expectation of the left and right terms, we get the same bound unconditionally. Applying the union bound,

$$\mathbb{P}\Big(\exists k \in [p],\ \widehat{I}_{m,k} \notin \big(I_k^0 - \varepsilon_m(u), I_k^1 + \varepsilon_m(u)\big)\Big) \leq p\exp\big[-c(m-1)\min\{u,u^2\}\big]. \tag{16}$$

It follows that $\widehat{I}_{m,k} \in \big(I_k^0 - \varepsilon_m(u), I_k^1 + \varepsilon_m(u)\big)$ for all $k \in [p]$, with probability at least $1 - 2p\exp[-\frac{1}{2}cm\min\{u,u^2\}]$, where we have used $\frac{1}{2}m \leq m - 1$ for $m \geq 2$. Take $u = \sqrt{2(c_1+1)\log p/(cm)}$ and assume that $u \leq 1$, so that $u^2$ is dominant in the bound. The proof is complete. $\qquad\square$

Combined with Lemma 2, we can transfer the results to $\widetilde{I}_{m,k} := \mathrm{imp}(\widetilde{y}_{[m]}^k)$.

**Corollary 2.** *The conclusion of Lemma 3 holds for $\widetilde{I}_{m,k}$ in place of $\widehat{I}_{m,k}$, with probability at least $1 - \check{p}^{-c} - 2e^{-\alpha m}$ for $\alpha = \log\frac{1}{8\delta}$.*

Note that for $\delta < 1/8$, the bound holds with high probability. Consider the hard sparsity case, where $\sigma_{f,k}^2 = 0$ for $k \notin S$, hence according to Corollary 2, $\widetilde{I}_{m,k} > I^0 - \varepsilon_m$ for $k \notin S$, where $I^0 := v_{f,*}^2 + \sigma_w^2$. On the other hand $\widetilde{I}_{m,k} < I_k^1 + \varepsilon_m$ for $k \in S$. Thus, as long as $I^0 - I_k^1 > 2\varepsilon_m$, the selection algorithm correctly favors the $k$th feature in $S$, over the inactive ones (recall that lower impurity is better). We have our main result after substituting $n/2$ for $m$:

**Theorem 2.** *Assume an additive model with* (F1)–(F2). *Let $\alpha = \log\frac{1}{8\delta}$ for $\delta \in (0, 1/8)$. The* DStump *algorithm, which selects the "s" least impure features at the root, succeeds in model selection, with probability at least $1 - \check{p}^{-c} - 2e^{-\alpha n/2}$ if $\log\check{p}/n \leq C_1$ and*

$$\min_{k \in S} g_{f,k}(\delta) \;\geq\; C(\sigma_{f,*}^2 + \sigma^2)\sqrt{\frac{\log\check{p}}{n}} \tag{17}$$

Note that $v_{f,*}^2$ does not directly appear in the result, only $\sigma_{f,*}^2$. Let us state the consequence of Theorem 2 for the special case of linear models, namely the case where $f_k(x) = \beta_k x$ for some coefficient vector $\beta = (\beta_k, k \in [p])$. Let $\kappa_1^2 := \frac{1}{12}$ be the variance of $U \sim \text{unif}(0,1)$ and $\kappa_2^2 := \|U\|_{\psi_2}^2$ (another universal constant). Then (F1) holds with $\sigma_{f,k}^2 = \kappa_2^2\beta_k^2$. Note also that $v_{f,k}^2 = \kappa_1^2\beta_k^2$. Thus, in this case $\sigma_{f,*}^2 \asymp v_{f,*}^2 \asymp \|\beta\|_2^2$.

Assumption (F2) holds trivially for any $\beta$ since it states $\alpha^2\kappa_1^2\beta_k^2 \leq (1-\delta)^2\kappa_1^2\beta_k^2$ for all $\alpha \leq 1 - \delta$. The gap in (2) reduces to $g_{f,k}(\delta) = \kappa_1^2[1 - (1-\delta)^2]$. Thus, we obtain the following corollary of Theorem 2.

**Corollary 3.** *Assume a linear (additive) model with $f_k(x) = \beta_k x$. Let $\alpha = \log\frac{1}{8\delta}$ and $\xi = 1 - (1-\delta)^2$ for $\delta \in (0, 1/8)$. The* DStump *algorithm, which selects the "s" least impure features at the root, succeeds in model selection, with probability at least $1 - \check{p}^{-c} - 2e^{-\alpha n/2}$ if $\log\check{p}/n \leq C_1$ and*

$$\min_{k \in S} \beta_k^2 \;\geq\; \frac{C}{\xi}(\|\beta\|_2^2 + \sigma^2)\sqrt{\frac{\log\check{p}}{n}} \tag{18}$$

The quantity $|\beta_S|_{\min}^2 := \min_{k \in S}\beta_k^2$ appearing in Corollary 3 is a well-known parameter controlling hardness of subset recovery. Although Corollary 3 applies to a general $\beta$, it is worthwhile to see its consequence in a special regime of interest where $|\beta_S|_{\min}^2 \asymp 1/s$, corresponding to $\|\beta\|_2 \asymp 1$. We get the following immediate corollary:

**Corollary 4.** *Assume $|\beta_S|_{\min}^2 \asymp 1/s$, $\sigma^2 \asymp 1$ and $\log\check{p}/n = O(1)$. Then* DStump *succeeds with high probability if $n \gtrsim s^2\log\check{p}$.*

The minimax optimal threshold for support recovery in the regime of Corollary 4 is known to be $n \asymp s\log\check{p}$ [Wai09b], and achieved by LASSO [Wai09a]. Although this result is obtained for Gaussian design, the same argument goes through for our uniform ensemble. Compared to the optimal threshold, using DStump we pay a small factor of $s$ in the sample complexity. However, DStump is not tied to the linear model and we expect that a similar performance can be extended to much more general nonlinear settings. In other words, we expect to pay a price due to flexibility of DStump, and not being tailored to the linear setting.

# 3 Class of valid $f$

Let us consider the class of functions $f : [0,1] \to \mathbb{R}$ that satisfy conditions (F1)–(F2). Since variance impurity is invariant to a shift, without loss of generality, we will assume $f(0) = 0$, and consider the class:

$$\mathcal{F} := \{f : [0,1] \to \mathbb{R} : \ f(0) = 0, \ \mathbb{E}[f(U)]^2 < \infty\}. \tag{19}$$

Note that $\mathbb{E}[f(U)]^2 = \alpha \mathbb{E}[f(\alpha U)]^2 + (1-\alpha)\mathbb{E}[f(1-\alpha U)]^2$ for all $\alpha \in [0,1]$. Then, for any $f \in \mathcal{F}$, we have $\mathbb{E}[f(\alpha U)]^2 < \infty$ for all $\alpha \in [0,1]$ (the case $\alpha = 0$ is trivial).

Condition (F1) is relatively mild and is satisfied by a large class of functions:

**Lemma 4.** *Let $f \in \mathcal{F}$. Then* (F1) *is satisfied if $f$ is Lipschitz or bounded.*

For a Lipschitz function, we have $|f(x) - f(y)| \le L_f |x - y|$ for all $x, y \in [0,1]$. Then, $|f(x)| = |f(x) - f(0)| \le L_f |x|$, hence

$$\|f(\alpha U)\|_{\psi_2}^2 \le \alpha^2 L_f^2 \|U\|_{\psi_2}^2 \le L_f^2 \|U\|_{\psi_2}^2, \quad \forall \alpha \in [0,1]$$

giving the desired uniform bound on the sub-Gaussian norm. For a bounded function $|f(x)| \le B$, $\forall x \in [0,1]$. Then $\|f(\alpha U)\|_{\psi_2}^2 \le B^2$ for all $\alpha \in [0,1]$ by (SG-3).

Condition (F2) is more stringent. A slightly stronger condition is that $\alpha \mapsto \mathrm{var}[f(\alpha U)]$ is nondecreasing on $[0,1]$. For example, we have:

**Lemma 5.** *Assume that $f$ can be extended to a continuously differentiable function on an open interval $I \supset [0,1]$, i.e., $f = F|_{[0,1]}$ where $F : I \to \mathbb{R}$ is continuously differentiable. Then* (F2) *holds if both $x \mapsto x f'(x)$ and $f$ are monotone of the same kind.*

*Proof.* Let us justify validity of interchanging differentiation (w.r.t. $\alpha$) and expectation, for example, for $\psi(\alpha) := [f(\alpha U)]^2$ where $\psi'(\alpha) = 2U f'(\alpha U) f(\alpha U)$. Then, $\psi(\alpha)$ is integrable for all $\alpha \in [0,1]$ (see the comment following definition of $\mathcal{F}$) and so is $\sup_{\alpha \in [0,1]} |\psi'(\alpha)|$ (since $x \mapsto |f'(x)f(x)|$ is continuous hence bounded on $[0,1]$). It follows that $\alpha \mapsto \mathbb{E}[f(\alpha U)]^2$ is continuously differentiable with derivative $\mathbb{E}\psi'(\alpha)$. A similar argument works for justifying differentiating $\mathbb{E}f(\alpha U)$ under expectation.

Letting $v(\alpha) := \mathrm{var}[f(\alpha U)] = \mathbb{E}[f(\alpha U)]^2 - [\mathbb{E}f(\alpha U)]^2$, it follows that

$$v'(\alpha) = 2\,\mathrm{cov}\left(U f'(\alpha U), \, f(\alpha U)\right), \quad \alpha \in (0,1).$$

Since the functions $x f'(x)$ and $f(x)$ are both monotone of the same kind, $\alpha U f'(\alpha U)$ and $f(\alpha U)$ are positively correlated according to Lemma 7, i.e. $v' \ge 0$ over $(0,1)$, hence $v$ is nondecreasing, implying (F2). $\square$

The conditions of Lemma 5 are satisfied for example for $C^1$ *convex nondecreasing* functions. To see this note that both $f$ and $f'$ are nondecreasing for such functions. Since $f(0) = 0$, we have $f \ge 0$, from which it follows that $f'(0) \ge 0$, hence $f' \ge 0$. Then, if $x_1 \le x_2$, both in $[0,1]$, we have $x_1 f'(x_1) \le x_1 f'(x_2) \le x_2 f'(x_2)$, where the second inequality uses $f' \ge 0$. Hence $x f'(x)$ is also nondecreasing, as desired. By noting that the variances of $f$ and $-f$ are equal, the same holds for $C^1$ *concave nonincreasing* functions.

Figure 1: Plot of the function in Lemma 6 (left) and the corresponding $\alpha \mapsto \mathrm{var}[f(\alpha U)]$ (right), for $a = \varepsilon = 10$. Note that the function violates (F2) over, say, $[0.2, 1]$.

However, conditions of the Lemma 5 could also hold for functions beyond those two categories. For example, consider $f(x) = (x + \varepsilon)^r$ for $r \in (0, 1)$ and $\varepsilon \in (0, 1)$. The function is concave and increasing on $[0, 1]$, however $xf'(x) = rx(x + \varepsilon)^{r-1}$ is also increasing, hence satisfies (F2) by Lemma 5.

On the other hand, it is not hard to construct functions that violate (F2). For example, consider the *wedge* $f(x) = \frac{1}{2} - |x - \frac{1}{2}|$ and let $v_f(\alpha) := \mathrm{var}[f(\alpha U)]$. Then, $v_f(\frac{1}{2}) = 1/48 > 11/576 = v_f(\frac{3}{4})$, so (F2) is violated for $\delta = 1/4$. It is also possible to construct a monotone function that violates (F2). For example, the following concave increasing function violates (F2) over any desired sub-interval $(\gamma, 1)$, for $\gamma > 0$, by taking $a$ and $\varepsilon$ large enough (cf. Figure 1):

**Lemma 6.** *Let $f(x) = \min\{ax, \frac{1}{a}(x + \varepsilon)\}$ for $a, \varepsilon > 0$, and assume $x_0 := \varepsilon/(a^2 - 1) \in (0, 1)$. Then, with $\rho := x_0/\alpha$, we have*

$$v_f(\alpha) := \mathrm{var}[f(\alpha U)] = \frac{x_0^2}{12}\left[a^2\rho + a^{-2}\rho^{-2}(1 - \rho)^3\right] + \frac{1}{4a^2}(\rho\varepsilon + x_0)^2(\rho^{-1} - 1), \quad \text{for } \rho \leq 1,$$

*and* $= \frac{x_0^2}{12}\rho^{-2}$ *for* $\rho > 1$.

*Proof.* Let $x_0 := \varepsilon/(a^2 - 1)$, the point where the two branches of $f$ switch. Assume $\alpha \geq x_0$. Take $U_1$ and $U_2$ to have uniform distributions on $(0, x_0/\alpha)$ and $(x_0/\alpha, 1)$. A mixture of these two distributions with weights $x_0/\alpha$ and $(1 - x_0/\alpha)$ is the $\mathrm{unif}(0, 1)$. Thus, letting $U = ZU_1 + (1 - Z)U_2$ where $Z \sim \mathrm{Ber}(x_0/\alpha)$ independent of $U_1$ and $U_2$, we have $U \sim \mathrm{unif}(0, 1)$. Note that given $Z = 1$, $\alpha U = \alpha U_1 \sim \mathrm{unif}(0, x_0)$. We have $\mathrm{var}[f(\alpha U)|Z = 1] = \mathrm{var}[a(\alpha U_1)] = \kappa_1^2 a^2 x_0^2$. Similarly, $\mathrm{var}[f(\alpha U)|Z = 0] = \mathrm{var}[\frac{1}{a}(\alpha U_2 + \varepsilon)] = \kappa_1^2(\alpha - x_0)^2/a^2$, hence with $\rho := x_0/\alpha$,

$$\mathbb{E}\,\mathrm{var}[f(\alpha U) \mid Z] = \kappa_1^2 x_0^2\left[a^2\rho + a^{-2}\rho^{-2}(1 - \rho)^3\right].$$

On the other hand $\mathbb{E}[f(\alpha U)|Z = 1] = \mathbb{E}[a(\alpha U_1)] = ax_0/2 =: A$ and $\mathbb{E}[f(\alpha U)|Z = 0] = \mathbb{E}[\frac{1}{a}(\alpha U_2 + \varepsilon)] = a^{-1}(\varepsilon + (x_0 + \alpha)/2) =: B$. Then,

$$\mathrm{var}\,\mathbb{E}[f(\alpha U) \mid Z] = \mathrm{var}[(A - B)Z] = \frac{1}{4a^2}[a^2 x_0 - (2\varepsilon + x_0 + \alpha)]^2\rho(1 - \rho)$$

$$= \frac{1}{4a^2}(\varepsilon + \alpha)^2\rho(1 - \rho).$$

The desired result follows from law of total variance. $\qquad\square$

Finally, let us comment on the gap condition. Using the notation $v_f(\alpha) := \mathrm{var}[f(\alpha U)]$, we have $g_f(\delta) = v_f(1) - v_f(1 - \delta)$. Thus, a sufficient condition for a positive gap is that $v_f$ be (strictly) increasing, which for example follows from Lemma 5 if the functions there are strictly monotone. Strict monotonicity of $v_f$ is however is a lot stronger than what we need. One just needs a positive gap at some point $1 - \delta \in (7/8, 1)$ for Theorem 2 to imply a sample complexity bound for recovery. The gap condition is in general weaker that (F2). In fact, for any sufficiently regular function $f$, first-order linear approximation of $f$ near $x = 1$ gives a positive gap as long as the slope of the line is nonzero. The larger the slope is (in absolute value), the larger the gap is and hence the easier it is to detect that $f$.

## 3.1 Technical lemmas

**Lemma 7.** *Let $f(\cdot)$ and $g(\cdot)$ be nondecreasing real-valued functions on $\mathcal{X}$, and for a random element $X \in \mathcal{X}$, let us write $f = f(X)$ and $g = g(X)$. Assume that $f$, $g$ and $fg$ are integrable. Then, $\mathrm{cov}(f, g) \geq 0$.*

*Proof.* Let $X'$ be an independent copy of $X$. Then, $[f(X) - f(X')][g(X) - g(X')] \geq 0$. Taking expectations gives the result. □

# 4 Details of correlated linear setting

We take the following approach to generalize our result to the correlated case: (1) We show a version of Theorem 1, which holds for an "approximately sparse" parameter $\widetilde{\beta}$ with uncorrelated design. (2) We derive conditions on $M$ such that the correlated case can be turned into the uncorrelated case with approximate sparsity. The following theorem details Step 1:

**Theorem 3.** *Assume Model 1(i)-(ii) with $M = I$, but instead of (iii) let $\beta = \widetilde{\beta}$, a general vector in $\mathbb{R}^p$. Let $S$ be any subset of $[p]$ of cardinality $s$. The DSTUMP algorithm, which selects the "s" least impure features at the root, recovers $S$, with probability at least $1 - \breve{p}^{-c} - 2e^{-\alpha n/2}$ if $\log \breve{p}/n \leq C_1$ and $\xi|\widetilde{\beta}_S|_{\min}^2 - \|\widetilde{\beta}_{S^c}\|_\infty^2 > C(\|\widetilde{\beta}\|_2^2 + \sigma^2)\sqrt{(\log \breve{p})/n}$.*

The theorem holds for any $\widetilde{\beta}$ and $S$, but the gap condition required is likely to be violated unless $\widetilde{\beta}$ is approximately sparse w.r.t. $S$. Going back to Model 1, we see that setting $\widetilde{\beta} = M\beta$ transforms the model with correlated design $X$, and exact sparsity on $\beta$, to the model with uncorrelated design $\widetilde{X}$, and approximate sparsity on $\widetilde{\beta}$. The following corollary gives sufficient conditions on $M$, so that Theorem 3 is applicable. Recall the usual (vector) $\ell_\infty$ norm, $\|x\|_\infty = \max_i |x_i|$, the matrix $\ell_\infty \to \ell_\infty$ operator norm $\|A\|_\infty = \max_i \sum_j |A_{ij}|$, and the $\ell_2 \to \ell_2$ operator norm $\|A\|_2$.

**Corollary 5.** *Consider a general ICA-type Model 1 with $\beta$ and $M$ satisfying*

$$\|\beta_S\|_\infty \leq \gamma|\beta_S|_{\min}, \quad \|M_{SS} - I\|_\infty \leq \frac{1 - \rho}{\gamma}, \quad \|M_{S^c S}\|_\infty \leq \frac{\rho}{\gamma}\sqrt{\xi(1 - \kappa)} \tag{20}$$

*for some $\rho, \kappa \in (0, 1]$ and $\gamma \geq 1$. Then, the conclusion of Theorem 3 holds, under the gap condition for the uncorrelated setting with $C/\xi$ replaced with $C\|M_{SS}\|_2^2/(\kappa \xi \rho^2)$.*

$C\|M_{SS}\|_2^2/(\kappa \xi \rho^2)$ plays the role of a new constant. There is a hard bound on how big $\xi$ can be, which via (20) controls how much correlation between off-support and on-support features

are tolerated. For example, taking $\delta = 1/9$, we have $\alpha = \log(9/8) \approx 0.1$, $\xi = 17/81 \approx 0.2$ and $\sqrt{\xi} \approx 0.45$ and this is about as big as it can get (the maximum we can allow is $\approx 0.48$). $\kappa$ can be arbitrarily close to 0, relaxing the assumption (20), at the expense of increasing the constant in the threshold. $\gamma$ controls deviation of $|\beta_j|, j \in S$ from uniform: in case of equal weights on the support, i.e., $|\beta_j| = 1/\sqrt{s}$ for $j \in S$, we have $\gamma = 1$. Theorem 1 for the uncorrelated design is recovered, by taking $\rho = \kappa = 1$.

*Proof of Theorem 3.* All the analysis of the linear case goes through with a general $\widetilde{\beta}$ and a general candidate support set $S$, providing concentration of the impurities around their mean. What will be different is the mean impurity values. Recall that $\widehat{I}_{m,k}$ is the impurity reduction of variable $k$, expected to be low on the candidate support $S$ and high off the support. We have support recovery if $\max_{k \in S} \widehat{I}_{m,k} < \min_{k \notin S} \widehat{I}_{m,k}$. Using Lemma 3, it is enough to have

$$\max_{k \in S} I_k^1 + \varepsilon_m \;\; < \;\; \min_{k \notin S} I_k^0 - \varepsilon_m \tag{21}$$

which in turn is equivalent to

$$2\varepsilon_m < \min_{k \notin S} I_k^0 - \max_{k \in S} I_k^1 = \kappa_1^2 \big[ \; \min_{k \notin S}(-\widetilde{\beta}_k^2) - \max_{k \in S}(-\xi\widetilde{\beta}_k^2) \; \big]$$
$$= \kappa_1^2 \big[ \; -\max_{k \notin S} \widetilde{\beta}_k^2 + \xi \min_{k \in S} \widetilde{\beta}_k^2 \; \big]$$

That is, we have support recovery w.h.p. if $2\kappa_1^{-2}\varepsilon_m < \xi|\widetilde{\beta}_S|_{\min}^2 - \|\widetilde{\beta}_{S^c}\|_\infty^2$ as desired. $\qquad\square$

*Proof of Corollary 5.* Under assumption (20),

$$\|\widetilde{\beta}_S - \beta_S\|_\infty \leq \|M_{SS} - I\|_\infty \|\beta_S\|_\infty \leq (1-\rho)|\beta_S|_{\min}.$$

Since $|\widetilde{\beta}_j| \geq |\beta_j| - |\widetilde{\beta}_j - \beta_j| \geq |\beta_j| - \|\widetilde{\beta}_S - \beta_S\|_\infty$ for all $j \in S$, we have

$$|\widetilde{\beta}_S|_{\min} \; \geq \; |\beta_S|_{\min} - \|\widetilde{\beta}_S - \beta_S\|_\infty \; \geq \; \rho|\beta_S|_{\min}$$

On the other hand $\|\widetilde{\beta}_{S^c}\|_\infty \leq \|M_{S^cS}\|_\infty \|\beta_S\|_\infty \leq \rho\sqrt{\xi(1-\kappa)}\,|\beta_S|_{\min}$. It follows that $\xi|\widetilde{\beta}_S|_{\min}^2 - \|\widetilde{\beta}_{S^c}\|_\infty^2 \geq \rho^2\xi\kappa|\widetilde{\beta}_S|_{\min}^2$. $\qquad\square$

## 5    Proof of Lemma 1 and 2

Calculating the density. Recall that the density of $(U_{(1)}, \ldots, U_{(n)})$ is constant and equal to $n!$ over $\{(u_1, \ldots, u_n) : \; 0 < u_1 < u_2 < \cdots < u_{n-1} < u_n < 1\}$. Letting $u_0 := 0$, we can write the density as $\check{f}(u) := n! \prod_{i=1}^n 1\{u_{i-1} < u_i\}$, over $[0,1]^n$. Here, $1\{\cdot\}$ is the indicator of a set. To simplify notation let us write $g_{k,j}(u) := \prod_{i=k}^j 1\{u_{i-1} < u_i\}$. First, let us find the density of $(U_{(1)}, \ldots, U_{(m)})$, where $m = n/2$, by induction. We would like to compute

$$\int_{[0,1]^m} \check{f}(u) \, du_{m+1} \ldots du_n = n! \, g_{1,m}(u) I(u) \tag{22}$$

where $I(u) := \int_{[0,1]^m} g_{m+1,n}(u) \, du_{m+1} \dots du_n$. We have

$$I(u) = \int_{[0,1]^{m-1}} g_{m+1,n-1}(u) \left[ \int_0^1 1\{u_{n-1} < u_n\} du_n \right] du_{m+1} \dots du_{n-1} \tag{23}$$

$$= \int_{[0,1]^{m-1}} g_{m+1,n-1}(u) \left[ 1 - u_{n-1} \right] du_{m+1} \dots du_{n-1}. \tag{24}$$

Proceeding by induction, assume that we have shown

$$I(u) = \int_{[0,1]^{m-j}} g_{m+1,n-j}(u) \frac{1}{j!} (1 - u_{n-j})^j \, du_{m+1} \dots du_{n-j}. \tag{25}$$

Then, we have

$$I(u) = \int_{[0,1]^{m-j-1}} g_{m+1,n-j-1}(u) \left[ \int_0^1 1\{u_{n-j-1} < u_{n-j}\} \frac{1}{j!} (1 - u_{n-j})^j du_{n-j} \right] du_{m+1} \dots du_{n-j-1}. \tag{26}$$

The integral inside brackets is $\int_{u_{n-j-1}}^1 \frac{1}{j!} (1-w)^j dw = \frac{1}{(j+1)!} (1 - u_{n-j-1})^{j+1}$. It follows from the induction that $I(u) = \frac{1}{m!} (1 - u_m)^m$. In other words, recalling $n = 2m$, we have shown that the density of $(U_{(1)}, \dots, U_{(m)})$ is

$$\frac{(2m)!}{m!} (1 - u_m)^m 1\{u_1 < u_2 < \dots < u_m\} \tag{27}$$

over $[0,1]^m$. To obtain the density of $\widetilde{U}$, we have to average the above over all possible $m!$ permutations, giving $\frac{(2m)!}{m!m!} (1 - \max\{u_1, \dots, u_m\})^m$ over $[0,1]^m$.

Obtaining the infinite mixture. Hereafter, let $u = (u_1, \dots, u_m)$. We also refer to a distribution and its density (w.r.t. Lebesgue measure) interchangeably. Ideally, we would like to approximate the distribution derived above, whose density we denote by $f(u) := \binom{2m}{m}(1 - \|u\|_\infty)^m$, with the distribution of an IID sequence of $\text{Unif}(0, 1-\delta)$ of length $m$. Unfortunately, this is not possible with vanishing error as $m \to \infty$. Instead, we take a clue from di Finetti theorem. Since $f$ is an exchangeable distribution, it can be written as a mixture of IID ones. Let us assume that we can approximate $f$ with a mixture of IID $\text{unif}(0, r)$, where $r$ varies according to some density $F$. A bit more precisely, noting that the density of an $m$-vector with IID $\text{Unif}(0, r)$ entries can be written as $u \mapsto r^{-m} 1\{\|u\|_\infty < r\}$, we approximate $f(u)$ with the following mixture density

$$h(u) := \int_0^1 \frac{1\{\|u\|_\infty < r\}}{r^m} F(r) dr. \tag{28}$$

Let us compute the total variation distance between distributions corresponding to $f$ and $h$. Equivalently, we look at the $L_1$ norm of $f - h$. Since $f - h$ is only a function of $\|u\|_\infty$, we can integrate by looking at shells of the $\ell_\infty$ ball of radius $t$ restricted to the nonnegative orthant. Note that $\text{vol}(t\mathbb{B}_\infty^m \cap \mathbb{R}_+^m) = t^m$, hence the volume of the corresponding shell is $d\,\text{vol}(t\mathbb{B}_\infty^m \cap \mathbb{R}_+^m) = mt^{m-1}dt$. It follows that

$$d_{\text{TV}}(f, h) = \int_{[0,1]^m} |f(u) - h(u)| du = \int_0^1 |\widetilde{f}(t) - \widetilde{h}(t)| m t^{m-1} dt \tag{29}$$

where

$$\widetilde{f}(t) = \binom{2m}{m}(1-t)^m, \quad \text{and} \quad \widetilde{h}(t) = \int_0^1 \frac{1\{t < r\}}{r^m} F(r) dr \tag{30}$$

Note that we can make the total variation vanish if we can choose $F$ so that $\widetilde{f} = \widetilde{h}$. Differentiating this identity, notating that $\widetilde{h}(t) = \int_t^1 r^{-m} F(r) dr$, and using fundamental theorem of calculus, we get

$$-m\binom{2m}{m}(1-t)^{m-1} = -t^{-m}F(t) \implies F(t) = m\binom{2m}{m}t^m(1-t)^{m-1} \tag{31}$$

We note that $m\binom{2m}{m} = 1/B(m, m+1) = 1/B(m+1, m)$. The above $F$ is in fact the density of the Beta distribution with parameters $m$ and $m+1$. Thus, we have shown that $f$ is a mixture of IID unif$(0, \gamma)$ $m$-vectors with mixing variable $\gamma \sim \text{Beta}(m, m+1)$, completing the proof of Lemma 1.

Approximation by truncating range. The last step is to show that we can replace the exact distribution $f$ with one that has strictly smaller range. Consider a mixture of the form

$$\widehat{h}_\delta(u) := \int_0^1 \frac{1\{\|u\|_\infty < r\}}{r^m} \widehat{F}_\delta(r) dr, \quad \widehat{F}_\delta(r) := \frac{1}{c(\delta)} F(r) 1\{r < 1 - \delta\} \tag{32}$$

where $F$ is as in (31) and $\delta > 0$. Here, $c(\delta) = \int_0^{1-\delta} F(r) dr$ is the normalizing constant. Let us also define the unnormalized version $\check{h}_\delta(u) := c(\delta) \widehat{h}_\delta(u)$. The total variation distance of $\check{h}_\delta$ with $f$ is

$$\|f - \check{h}_\delta\|_{L^1(\mathbb{R}^m)} = \int_0^1 \left| \int_0^1 \frac{1\{t < r\}}{r^m} F(r) dr - \int_0^{1-\delta} \frac{1\{t < r\}}{r^m} F(r) dr \right| mt^{m-1} dt$$

$$= \int_0^1 \int_{1-\delta}^1 \frac{1\{t < r\}}{r^m} F(r) dr \, mt^{m-1} dt$$

$$= \int_{1-\delta}^1 \left[ \int_0^1 1\{t < r\} mt^{m-1} dt \right] \frac{F(r)}{r^m} dr,$$

by Fubini theorem. The inner integral is $\int_0^r mt^{m-1} dt = r^m$. It follows that $\|f - \check{h}_\delta\|_{L^1(\mathbb{R}^m)} = \int_{1-\delta}^1 F(r) dr = 1 - c(\delta)$. On the other hand, $\|\widehat{h}_\delta - \check{h}_\delta\|_{L^1(\mathbb{R}^m)} = (1 - c(\delta))\|\widehat{h}_\delta\|_{L^1(\mathbb{R}^m)} = 1 - c(\delta)$, since $\widehat{h}_\delta$ is a normalized density. Hence, $\|f - \widehat{h}_\delta\|_{L^1(\mathbb{R}^m)} \leq 2[1 - c(\delta)]$. It remains to bound $1 - c(\delta)$. We have

$$1 - c(\delta) = \int_{1-\delta}^1 F(r) dr = m\binom{2m}{m} \int_{1-\delta}^1 r^m(1-r)^{m-1}. \tag{33}$$

Using $r^m(1-r)^{m-1} \leq \delta^{m-1}$ over $(1-\delta, 1)$, and $\binom{n}{k} \leq (en/k)^k$, we have

$$1 - c(\delta) \leq m\binom{2m}{m}\delta^m \leq m\left(\frac{2me}{m}\right)^m \delta^m = m(2e\delta)^m \leq (8\delta)^m \tag{34}$$

(We can replace 8 with $2e^{1/e+1} \approx 7.86$.) Taking $\delta = e^{-\alpha}/8$ for $\alpha > 0$, we have $1 - c(\delta) \leq \exp(-\alpha m)$. The proof is complete.

Figure 2: Support recovery performance in a linear regression model augmented with possible nonlinearities for $n = 128$. (a) Linear case with uncorrelated design. (b) Linear case with correlated design. (c) Nonlinear additive model with exponentials of covariates and uncorrelated design. (d) Nonlinear model with interaction terms and uncorrelated design. (e) Nonlinear additive model with exponentials of covariates, interaction terms, and uncorrelated design. (f) Nonlinear additive model with exponentials of covariates, interaction terms, and correlated design.

# 6   Additional Simulation Results

In this section we present simulation results to supplement the results from the main text. First, we replicated the simulations for different sample sizes, i.e. $n = 128$ and $n = 10,000$. The results are presented in Figure 2 and Figure 3. We observe that all methods are sensitive to small sample sizes, with more complex algorithms like SPAM more prone to failure. In the presence of sufficiently large sample sizes, we can observe the asymptotic efficiency of different methods. Under a correlated design, DSTUMP and SIS close the gap with Lasso and SPAM, while TREEWEIGHT shows inferior performance. In contrast, TREEWEIGHT widens its margin over other methods in settings with interaction terms.

The results presented thus far evaluate various methods based on the fraction of support recovered. We next report results under the same modeling settings, but using a different evaluation metric. Specifically, we report the probability of exact support recovery. As illustrated in Figure 4, this metric shows qualitatively similar behavior as our previous experimental results, though the performance of each method degrades much more drastically, as this metric is more stringent than our initial evaluation metric.

Finally, recall that our reported results are averages over 100 replicates. Here we present the results from Figure 1 in the main text with 95% confidence intervals in order to demonstrate the statistical significance of these results. The results are shown in Figure 5, where dashed

Figure 3: Support recovery performance in a linear regression model augmented with possible nonlinearities for $n = 10000$. (a) Linear case with uncorrelated design. (b) Linear case with correlated design. (c) Nonlinear additive model with exponentials of covariates and uncorrelated design. (d) Nonlinear model with interaction terms and uncorrelated design. (e) Nonlinear additive model with exponentials of covariates, interaction terms, and uncorrelated design. (f) Nonlinear additive model with exponentials of covariates, interaction terms, and correlated design.

lines represent the confidence end points.

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

Figure 4: The probability of full support Support recovery in a linear regression model augmented with possible nonlinearities for $n = 1024$. (a) Linear case with uncorrelated design. (b) Linear case with correlated design. (c) Nonlinear additive model with exponentials of covariates and uncorrelated design. (d) Nonlinear model with interaction terms and uncorrelated design. (e) Nonlinear additive model with exponentials of covariates, interaction terms, and uncorrelated design. (f) Nonlinear additive model with exponentials of covariates, interaction terms, and correlated design.

Figure 5: Support recovery performance in a linear regression model augmented with possible nonlinearities for $n = 1024$. Dashed lines represent the 95% confidence intervals. (a) Linear case with uncorrelated design. (b) Linear case with correlated design. (c) Nonlinear additive model with exponentials of covariates and uncorrelated design. (d) Nonlinear model with interaction terms and uncorrelated design. (e) Nonlinear additive model with exponentials of covariates, interaction terms, and uncorrelated design. (f) Nonlinear additive model with exponentials of covariates, interaction terms, and correlated design.