[Reviews · NeurIPS 2017]

Reviewer 1



The article tackles the problem of variable importance in regression trees. The strategy is to select the variables based on the impurity reduction they induce on label Y. The main feature of this strategy is that the impurity reduction measure is based on the ordering of Y according to the ranking of the X variable under consideration, therefore it measures the relationship between Y and any variable in a more robust way than simple correlation would. The authors prove that this strategy is consistent (i.e. the true explanatory variables are selected) in a range of settings. This is then illustrated on a simulated example where the results displayed are somewhat the ones one could have expected: the proposed procedure is able to account for monotone but non linear relationships between X and Y so it yields better results than simple correlations. In the same time, it cannot efficiently deal with interaction since explanatory variables are considered one by one - compared with procedures that consider a complete tree rather than only a stump. Overall the idea that is presented is very simple but as mentioned by the authors the theoretical analysis of the strategy is not completely trivial. Although there are some weaknesses in the manuscript the ideas are well presented and the sketchs of proof presented in section 3.1 and 3.2 provide a good lanscape of the complete proof. However there are some weaknesses in the Simulation section, and it is a bit surprising (and disappointing) that there is no discussion section at all. Regarding the Simulation section, first it seems that the quality criterion that is represented is actually not the criterion under study in section 2 and 3. The theoretical results state the consistency of the procedure, i.e. the probability that the complete set of s predictors is retrieved, so the authors could have displayed the proportion of runs where the set is correctly found. Instead, they display the fraction of the support that is recovered, and it is not clear whether this is done on a single simulated dataset or averaged. If only on a single simulation then it would be more relevant to repeat the experiment on several runs to have a clearer picture of what happens. Another remark is that the authors mention that they consider the model Y=X\beta +f(X_S) + w, but do they consider this model both to generate the data and perform the analysis, or just for the generation and then a purely linear model is used for analysis ? It seems that the second hypothesis is the good one but this should be clearly mentioned. Also it would be nice to have on an example the histogram of the criterion that is optimized by each procedure (when possible) to empirically evaluate the difficulty of the problem. Regarding the proof itself, some parts of it should be more commented. For instance, the authors derive the exact distribution of \tilde{U}, but then they use a truncated version of it. It may seem weird to approximate the (known) true distribution to derive the final result on this distribution. The authors should motivate this step and also provide some insight about what is lost due to this approximation. Also, is there any chance that the results presented here can be easily extended to the classification case ? And lastly, since the number of active feature is unknown in practice, is there any way the strategy presented here could help estimating this quantity (also I understand it is not the core of the article) ?

Reviewer 2



This paper studies the variable selection consistency of decision trees. A new algorithm called "DStump" is proposed and analyzed which simply checks the variable importance via comparing the corresponding impurity reduction at the root node. DStump needs to know the correct number of true features to work. For both the linear models with uncorrelated design and the general additive nonlinear models, the model selection consistency is proved to hold true with overwhelming probability. The analysis is further generalized to linear correlated design where the gap of signal strength is of course required. The analysis with a truncated beta distribution is inspiring and promising. The results on decision tree are new (to the best of my knowledge). The paper is written with a good style and the literature review is also sufficient. Because of the time restriction, I have not gone through the math details in the attached technical report. Some minor points: 1. The authors are encouraged to add a short paragraph before Algorithm1 to define the problem. 2. In lines 240-241, while all the y's are just real numbers, how can they be regarded as drawn from Uniform[0,1/2]?

Reviewer 3



This paper deals with the problem of variable selection using decision tree. More precisely,the authors provide a new algorithm called DStump which relies on the impurity-based information to perform variable selection. They provide theoretical guarantees for the proposed method. I find the paper interesting and very clear. The problem of variable selection using decision tree is not new but there is a lack of theoretical analysis. Hence, from a theoretical point of view, I think that this paper provide a relevant contribution. Minor comment: Why do you not use the permutation-based importance method (used in random forests) in the simulations ?